

# Detection of deterministic and probabilistic convective initiation using Himawari-8 Advanced Himawari Imager data

Sanggyun Lee[1], Hyangsun Han[2], Jungho Im[1,3], Eunna Jang[1], and Myong-In Lee[1]

[1]School of Urban and Environmental Engineering, Ulsan National Institute of Science and Technology (UNIST), Ulsan, 44949, South Korea
[2]Unit of Arctic Sea-Ice prediction, Korea Polar Research Institute, Incheon, 21990, South Korea
[3]Environmental Resource Engineering, State University of New York, College of Environmental Science and Forestry, 13210, Syracuse, NY, USA

*Correspondence to*: Jungho Im (ersgis@unist.ac.kr)

**Abstract.** Detection of Convective Initiation (CI) is very important because convective clouds bring heavy rainfall and thunderstorms that typically cause severe socioeconomic damages. In this study, deterministic and probabilistic CI detection models based on decision trees (DT), random forest (RF), and logistic regression (LR) were developed using Himawari-8 Advanced Himawari Imager (AHI) data obtained from June to August 2015 over the Korean Peninsula. We used a total of 12 interest fields that contain brightness temperature, spectral differences of the brightness temperatures, and their time trends to develop CI detection models. While the interest field of 11.2 μm $T_B$ was considered the most crucial to detect CI in the deterministic models and the probabilistic RF model, the trispectral difference, i.e., (8.6-11.2 μm) - (11.2-12.4 μm), was determined as the most important one in the LR model. The performance of the four models varied by CI case and validation data. Nonetheless, the DT model typically showed higher POD, while the LR model produced higher OA and CSI and lower FAR than the other models. The CI detection of the mean lead times by the four models were in the range of 32 - 35 min, which implies that convective clouds can be detected in 30 min advance before precipitation intensity exceeds 35 dBZ over the Korean Peninsula in summer using the Himawari-8 AHI data is possible.

## 1 Introduction

Atmospheric deep moist convection initiates shallow cumulus clouds, which may continue to grow vertically as cumulonimbus clouds, and this process is so called the convective initiation (CI) (Banacos et al., 2005; Bluestein et al., 1990; Weckwerth and Parsons, 2006). The moist convection appears in a variety of horizontal scales ranging from 1-10 km as individual convective clouds to ~1000 km as mesoscale convective systems (Houze Jr., 2004; Roberts and Lean, 1998; Miyamoto et al., 2013) with heavy rainfall and thunderstorm events (Amorati et al., 2000; Sieglaff et al., 2011; Zuidema, 2003; Haile et al., 2010; Hane et al., 2002; Vondou et al., 2010). The convective events in Northeast Asia often occur during the summer season accompanied by many meteorological hazards such as lightning, floods, and strong winds (Kim and Lee, 2006; Wang et al., 2004). These





hazards destroy infrastructure in the region and result in huge economic losses. Therefore, it has been desired to forecast CI in Northeast Asia with high accuracy in order to prevent socioeconomic damages caused by the convective events.

The decrease of atmospheric stability drives CI, which is attributed to various weather systems such as large-scale monsoonal fronts, the migration of frontal cyclones, and mesoscale convective systems (Craven et al., 2002; Houze Jr., 2004; Mecikalski

and Bedka, 2006). Although such unstable weather systems can increase the potential risk of CI over vast area, they actually trigger CI occupying much smaller area, making it difficult to predict the exact location. CI is characterized by the rapid variation of temperature and the increase of cloud-tops, which can be effectively measured by the brightness temperature ($T_B$) changes at multispectral channels including visible and infrared (IR) (Mecikalski and Bedka, 2006; Mecikalski et al., 2010). Geostationary satellites carry optical sensors that scan over a few thousand square kilometers with high temporal resolution (~

minutes) in the multispectral channels. Therefore, these geostationary satellites can be extremely useful in CI nowcasting. Previous studies developed CI nowcasting algorithms for geostationary satellites by determining a threshold or a range of values of $T_B$ at specific channels, and their spectral and/or temporal differences (Mecikalski and Bedka, 2006; Mecikalski et al., 2008; Walker et al., 2012; Morel and Senesi, 2002; Jewett et al., 2013; Merk and Zinner, 2013; Siewert et al., 2010; Sobajima, 2012; Han et al., 2015). Geostationary Operational Environmental Satellite (GOES) systems and Meteorological

Second Generation (MSG) are the representative geostationary satellites operated at the National Oceanic and Atmospheric Administration (NOAA) and European Organization for the Exploitation of Meteorological Satellites (EUMESAT), respectively. These two satellites have forecasted CI using their operational algorithms, i.e., SATellite Convection Analysis and Tracking (SATCAST) and Rapid Development Thunderstorms (RDT), respectively, which are basically based on empirical determination of the thresholds of interest fields in terms of CI development (Mecikalski and Bedka, 2006; Walker

et al., 2012; Morel and Senesi, 2002). These algorithms were assessed for CI cases in North America and showed a probability of detection (POD) over 0.8 (80%) and a false alarm rate (FAR) around 0.6 (60%). However, these algorithms for CI detection have not been validated over Northeast Asia.

Several algorithms for detecting CI over Northeast Asia were developed for the Multi-functional Transport SATellite-2 (MTSAT-2) geostationary satellite operated by Japan Meteorological Agency (JMA) and the Communication, Ocean, and

25 Meteorological Satellite (COMS) by Korea Meteorological Administration (KMA). The main instrument of MTSAT-2 is Imager which is composed of a total of 5 channels: a visible channel with 1 km-spatial resolution and 4 infrared channels with 4 km spatial resolution. JMA have developed the algorithm for the detection of CI using MTSAT-2 Imager over Japan area i.e., the Rapidly Developing Cumulus Areas (RDCA) derivation algorithm (Sobajima, 2012). The RDCA algorithm detects CI using several interest fields, but it was validated only during the summer season in 2011 (Sobajima, 2012). The performance

of the MTSAT-2 RDCA algorithm needs to be evaluated in different times and circumstances. Han et al. (2015) developed CI detection algorithms for COMS Meteorological Imager (MI) data by determining new rules and thresholds for the interest fields used in the RDCA through machine learning, based on the fact that the characteristics of the spectral channels of COMS MI is similar to those of MTSAT-2. The algorithms were validated for various CI cases over Korea, demonstrating good



performance with a POD as high as 75.5% and a FAR as low as 46.2%. Himawari-8, launched on 7 October 2014, is one of the geostationary satellites operated by JMA. The primary payload of Himawari-8 is the Advanced Himawari Imager (AHI) which captures the Asia-Pacific region using a 16-channel multispectral imager composed of 3 visible (blue, green, and red) and 13 IR channels with a spatial resolution of 0.5–2 km depending on the spectral channel (Bessho et al., 2016). Himawari-8

AHI scans full disk, the whole Earth as seen from the satellite (11000 × 11000 km), every 10 min. The scan interval for the full disk of Himawari-8 AHI is much shorter than that of MTSAT Imager (60 min) and COMS MI (180 min). Furthermore, the spatial and spectral resolutions of Himawari-8 AHI have been substantially improved in comparison to its predecessors. Therefore, Himawari-8 AHI can help enhance the performance of CI detection in Northeast Asia. However, there is no available algorithm for CI detection for Himawari-8 AHI so far. The spectral characteristics of Himawari-8 AHI are

comparable to the Advanced Baseline Imager (ABI) integrated onto the Geostationary Operational Environmental Satellite-R series (GOES-R) satellites (Schmit et al., 2005), a series of geostationary satellites of which the first one will be launched in November 2016 and operated by National Oceanic and Atmospheric Administration (NOAA). University of Alabama in Huntsville with NOAA developed a CI detection algorithm for GOES-R using 12 interest fields designed for the spectral bands of ABI (Walker and Mecikalski, 2011; Walker et al,. 2012; Melcikalski et al., 2015). The interest fields of the GOES-R CI

algorithm can be directly adopted for Himawari-8 AHI. However, the critical threshold values of the interest fields were empirically determined based on simulations and have not been fully assessed due to the lack of ABI-class datasets. This implies that the criteria of the interest fields should be optimized for use with Himawari-8 AHI data.

The existing literature on nowcasting CI commonly used deterministic approaches which classify clouds into CI and non-CI using the criteria of several interest fields based on simple thresholding approaches (Mecikalski and Bedka, 2006; Mecikalski

et al., 2008; Mecikalski et al., 2010; Morel and Senesi, 2002; Sieglaff et al., 2011; Roberts and Rutledge, 2003; Sobajima, 2012). Such deterministic approaches might provide incorrect classification results for unsampled pixels (or objects), especially around the boundaries of CI, increasing FAR of predictions. Probabilistic approaches produce significantly lower FAR than the deterministic ones through the selection of an appropriate probability threshold (Mecikalski et al., 2015). Mecikalski et al. (2015) developed CI nowcasting algorithms combining the interest fields derived from GOES and numerical

weather prediction (NWP) model data based on probabilistic approaches. They validated the performance of the probabilistic algorithms for CI cases in the United States, resulting in FAR of 10-18%, which is much lower than the existing deterministic CI detection algorithms for GOES (FAR ~48–60%) (Walker et al., 2012). However, Mecikalski et al. (2015) used fewer satellite-based interest fields than the GOES-R CI algorithm due to the limited number of spectral channels of GOES, which implies that it is unknown if such probabilistic approaches would work for Himawari-8 AHI as well.

In this study, CI detection algorithms for Himawari-8 AHI are developed and validated over the Korean Peninsula in East Asia. The objectives of this research were to (1) develop deterministic and probabilistic CI detection algorithms for Himawari-8 AHI data based on rule-based decision trees and random forest approaches and a logistic regression modelling technique, (2) evaluate the CI detection models in terms of performance and efficiency, (3) assess the strengths and weaknesses of the deterministic and probabilistic CI detection models based on CI cases and validation datasets, and (4) examine key predictor



variables for CI detection. This study extends our previous research in Han et al. (2015), where the COMS MI data were used. One of the main limitations of using COMS MI data in the previous study is its relatively coarse spatial resolution (4km), which is not enough to detect small convective clouds. The himawari-8 AHI used in the present study has higher spatial resolution of 2 km for IR channels. In addition, the higher spectral (i.e., 16 channels) and temporal (i.e., 10 min) resolutions

of AHI than COMS MI can significantly improve the forecast skill. The present research proposes not only deterministic approaches but also probabilistic ones for CI detection using Himawari-8 AHI data. Consequently, the use of advanced geostationary satellite data and various modelling techniques is expected to produce better CI forecast performance.

## 2 Data

### 2.1 Himawari-8 AHI

The specifications of Himawari-8 AHI are summarized in Table 1. Himawari-8 AHI scans three levels of regions: full disk, Japan area, and target area. Full disk images are acquired over the whole Earth as seen from the satellite every 10 min. Japan area images, collected every 2.5 min, cover the northeastern and southwestern areas of Japan (3,000 × 3,000 km). Himawari-8 AHI also scans a target area of 1,000 × 1,000 km every 2.5 min, which is dedicated to monitoring high impact meteorological

events. In this study, the full disk images were used even though Himawari-8 AHI scans the Japan area four times more often. This is because the Japan Area images do not cover the upstream side of storm developments in the East China Sea and western part of the Yellow Sea, which are important areas to forecast CI over the Korean Peninsula and Japan. The full disk images obtained for 10 cases of CI from June to August 2015 (Table 2) were used to develop and validate the deterministic and probabilistic CI detection models.

### 2.2 Weather radar echo and lightning data

Rainfall with ≥35 dBZ precipitation intensity measured by weather radar has been known to have significant correlation with the eventual development of cumulonimbus clouds (Mecikalski and Bedka, 2006; Mueller et al., 2003). Therefore, the threshold of ≥35 dBZ precipitation intensity has been widely used as the definition of convective events (Mecikalski and Bedka, 2006; Mecikalski et al., 2008, 2010; Roberts and Rutledge, 2003; Walker et al., 2012). In this study, the first occurrence of

rainfall with ≥35 dBZ precipitation intensity was defined as CI. KMA has operated a total of 10 weather radars in South Korea. They have produced Plan Position Indicator (PPI) in which the precipitation echoes measured at a given elevation angle are projected on a plane every 10 min and Constant Altitude PPI (CAPPI) images, which are calculated using several PPI elevations. A 1.5 km CAPPI with mosaic image was used to determine the areas and time of CI occurrences. Since the effective radius of the 1.5 km CAPPI is about 100 km, only the 1.5 km CAPPI echo at each radar within 100 km were used.





Lightning observation data was used as supplementary data for validating the CI detection results, especially for ocean areas. The lightning data were provided by KMA, which operates a ground-based Total Lightning Detection System (TLDS) since 2001. The TLDS has an average accuracy of 90% for lightning detection, with locational accuracy of 500 m over the land and 2 km over the ocean (Kar and Ha, 2003). The TLDS lightning observation data during the same period were used with the ground radar measurements for the validation of the CI detection models.

## 3 Methods to detect CI using geostationary satellite data

Because of the large similarity in spectral channels between Himawari-8 AHI and GOES-R ABI, one may consider to adopt the GOES-R CI algorithm directly to develop CI algorithms for Himawari-8 AHI. However, as mentioned, the GOES-R CI algorithm uses simple threshold values associated with the interest fields and the values were determined through many experimental simulations in a subjective way. The interest fields and threshold values have not been validated for Himawari-8 AHI. In order to develop more objective CI models for Himawari-8 AHI, rule-based decision trees and random forest machine learning approaches were used as well as a logistic regression model. The interest fields identified in the GOES-R CI algorithm were used as predictor variables in both deterministic and probabilistic approaches. The processing flow diagram of the proposed CI detection approaches is shown in Fig. 1.

### 3.1 Interest fields of GOES-R ABI CI algorithm

The interest fields of the GOES-R ABI CI algorithm (Table 3) were used as a set of predictors to develop the deterministic and probabilistic CI detection models for Himawari-8 AHI. All the interest fields are calculated only from IR channels in order to predict CI using both daytime and nighttime images. $T_B$ measured at 11.2 μm and its time trend represent cloud-top temperature and cloud-top cooling rate, respectively (Mecikalsk and Bedka, 2006; Mecikalski et al., 2010; Walker and Mecikalski, 2011). The interest fields from spectral differences provide information on cloud-top height (cloud depth) and glaciation at the time of image, while those from temporal differences provide information on the rate of vertical cloud-top growth.

Prior to the calculation of the interest fields, cloudless and cirrus regions were masked out from Himawari-8 AHI images using the criteria of the $T_B$ at 11.2 μm-channel < 288.5 K, -3 K < 12.4-11.2 μm-channels difference < 3 K, and -3 K < the trispectral difference < 3 K. Such criteria have been empirically used to identify clear sky and thin clouds in summer by KMA. The first criterion was used to remove land surface and cirrus from AHI images, while the others were employed to remove clear sky areas. To determine the time-dependent interest fields, it is essential to track the motion of cloud objects. A simple temporal overlap object tracking method (Walker and Mecikalski, 2011) was adopted in this study, which uses two consecutively obtained AHI images in order to track cloud objects. This object tracking method has a weakness that fast moving cloud objects



of small size might not be traced. However, the frequent scanning interval and fine spatial resolution of Himawari-8 AHI help mitigate the weakness of the tracking method. The motion of cloud objects was traced from two consecutive 11.2 μm-channel images of AHI and then the time-dependent interest fields were calculated.

**3.2 Deterministic and probabilistic approaches for CI detection**

The event of CI (i.e., CI vs. non-CI) is used as a dependent (i.e., response) variable in the CI detection models based on deterministic and probabilistic approaches. For seven Himawari-8 AHI image data including CI events over the Korean Peninsula, the pixels within cloud objects corresponding to the first occurrence of ≥35 dBZ precipitation intensity were extracted and considered as convective clouds (CI areas). The other clouds were identified as non-CI regions. In order to collect interest field samples to develop CI detection models, each CI and non-CI area was tracked through visual interpretation of

the 11.2 μm-channel images obtained 10–60 min before the first occurrence of ≥35 dBZ precipitation intensity, rather than using the temporal overlap object tracking method (Zinner et al., 2008) that was used to calculate the time-dependent interest fields. A total of 3,204 CI reference data (1,324 CI and 1,880 non-CI samples) were extracted from the AHI images and used to train and validate the deterministic and probabilistic CI detection models. From eight percent of the total images (42 scenes), 1,060 CI samples and 1,504 non-CI samples were extracted and used as training data, while 264 CI samples and 376 non-CI

samples extracted from the remaining AHI images (18 scenes) were used to validate the models. The models were further validated using 3 additional CI events, which were not used to extract samples.

In this study, three approaches including decision trees (DT), random forest (RF), and logistic regression (LR) were used for the development of CI detection models. DT and LR were used for deterministic and probabilistic CI detection, respectively. Meanwhile, RF was used for both deterministic and probabilistic detection of CI. DT has been widely used for classification

and regression tasks in the remote sensing field (Li et al., 2013; Lu et al., 2014; Kim et al., 2014; Torbick and Corbiere, 2015). See5, developed by RuleQuest Research, Inc. (Quinlan, 2015), was employed to perform the DT-based classification of clouds into CI and non-CI. See5 works by repeatedly splitting samples into two groups of greater homogeneity using an entropy-based parameter to generate a tree (Quinlan, 2015; Jensen and Im, 2007). Pruning to avoid overfitting is often applied when a decision tree is generated. One of the merits of See5.0 compared to other DT algorithms is that the generated tree can be

reproduced with multiple if-then rules, which makes it easier to interpret the results than the original tree (Im et al., 2008; Jensen et al., 2008; Rhee et al., 2008; Im et al., 2012; Kim et al., 2015).

RF uses a bootstrapping strategy from the original training data to produce a series of Classification and Regression Trees (CART) that is a non-parametric decision tree, which produces either classification or regression trees depending on whether the dependent variable is categorical or numerical (i.e., continuous) (Breiman, 2001). The numerous independent trees (e.g.,

500, 1000) are grown based on two randomizations including (1) a randomly selected subset of the training samples for each tree and (2) a randomly selected subset of input variables at each node of the tree. This way, RF overcomes the well-known limitation of CART that results are sensitive to the configuration and quality of training data (Lawrence and Wright, 2001;



Rhee et al., 2014). Thus, RF has recently gained popularity in remote sensing classification and regression (Kim et al., 2014; Li et al., 2014; Liu et al., 2015; Lu et al., 2013; Park et al., 2016; Yoo et al., 2012). Two approaches are generally adopted to reach a final conclusion from the independent decision trees including a simple majority voting and weighted majority voting strategy for classification, while a final value is either simply averaged from the results of the multiple regression trees or

averaged with weights for regression. In the probabilistic RF, the probability of an event occurrence is calculated using the ratio of voting for CI and Non-CI cases with 500 trees.

LR is one of the statistical regression methods that are used for modelling a categorical dependent variable using independent variables (Hosmer and Lemeshow, 2000). In this study, binary LR, a type of LR technique that deals with only two values for a dependent variable, was used to estimate the probability of CI occurrence. The logistic function is given as follows:

$$E(Y) = \frac{1}{1 + \exp\left[ -\left( \beta_0 + \sum_{j=1}^{k} \beta_j X_j \right) \right]} \qquad (1)$$

where $E$ is the expected value of the dependent variable $Y$, $k$ is the number of independent variables and $X_j$ is the value of the $j$ th independent variable, $\beta_0$ is the intercept from the linear regression equation, and $\beta_j$ is the weighting coefficient for the $j$ th independent variable. The logistic function produces $E$ values of [0, 1] which are used as the probability of whether CI will occur or not. DT, RF, and LR provide the relative importance of input variables when developing

models such as attribute usage, mean decrease accuracy, and the absolute value of weighting coefficients, respectively. See 5 produces the information on how frequently each variable is used in the results. RF calculates the decrease in accuracy of the model using out-of-bag data through the random permutation of a variable. Therefore, a higher mean decrease accuracy of a variable indicates more contribution of the variable to develop a model. In the LR, the exponentiation of weighting coefficients of the independent variables reflects the relative importance of the variables, which refers to the changes in the odds ratio

attributed to an input variable.

A series of typical accuracy metrics were calculated through confusion matrices to assess the performance of the CI detection models, including producer's and user's accuracies, overall accuracy, and kappa coefficients. In addition, the prediction results of the models were further assessed for the 3 cases of CI events over the Korean Peninsula by computing POD, FAR, overall accuracy (OA), and Critical Success Index (CSI) reflecting effects of both POD and FAR as follows (Mecikalski et al., 2015):

POD = A/(A+B)                     (2)

FAR = C/(A+C)                     (3)

OA = (A+D)/(A+B+C+D)              (4)

CSI = (A)/(A+B+C)                 (5)

where A is the number of CI objects that correctly detected as CI (i.e., hits), B is the number of CI objects incorrectly classified

as non-CI (i.e., misses), C is the number of non-CI objects incorrectly identified as CI (i.e., false alarm), and D indicates correct



negatives. A through D were counted from the results of the CI detection models using the Himawari-8 AHI images obtained 0–10, 10–20, 20–30, 30–40, and 40–50 min before the first occurrence of ≥35 dBZ intensity from the weather radar data based on CI objects. In order to identify CI objects in the AHI images that were obtained 0-50 min before the CI event occurred, distances from cloud objects to the location of the CI occurrence (i.e., ≥35 dBZ precipitation intensity) were calculated using

5    the atmospheric motion vector (AMV) product hourly generated from COMS MI (as Himawari-8 AHI does not provide AMV yet), assuming that the velocity and direction of moving clouds are constant over 1 h. Mean velocities and directions of the AMV of each cloud object for each case day were used to identify an overall motion vector. The cloud objects with a given direction within a given distance from the location of the first occurrence of ≥35 dBZ precipitation intensity were considered as CI. Overall POD, FAR, OA, and CSI for each CI detection model were computed based on the A-D values of three case

10   days.

For the 3 case days, the lead time for CI detection was calculated using a weighted mean depending on A (i.e., Hits) detected from the AHI images obtained before precipitation intensity exceeds 35 dBZ as follows (Han et al., 2015):

$$\frac{\sum A_t \times n}{\sum n} \quad (t = 0, 10, 20, 30, 40, 50, 60 \text{ min}) \tag{6}$$

where $A_t$ is the number of A counted from the AHI images obtained $t$ minutes before the first occurrence of ≥35 dBZ intensity

and $n$ is the number of $A_t$. The lead time of each CI detection model was determined using the $A_t$ and $n$ of all case days.

## 4 Performance and validation of CI detection models

### 4.1 Performance of CI detection models

The box plots of the 12 interest fields generated using CI and Non-CI reference data are depicted in Fig. 2. A line inside a coloured box indicates the median value of the data. The height of the coloured boxes represents the interquartile range of the

data, and the 1.5 times interquartile range is shown with the vertical centre lines. The dots above and below the vertical lines are outliers. The p-values in the upper left corner of each box plot were derived from t-test at the 95% confidence level. All the interest fields except for 6.2 - 7.3 μm time trend $T_b$ show p-values below 0.05. In particular, the interest field of 11.2, 6.2 - 11.2, and 6.2 - 7.3 μm $T_B$ showed noticeably low p-values (< 0.001), of which the interquartile ranges of CI and non-CI samples do not overlap each other.

The performances of the four CI detection models were assessed using confusion matrices produced from the test dataset (Tables 4-7). Since the probabilistic models produce the possibility of CI occurrence ranging from 0 to 100 %, the pixels with a possibility higher than 50 % were regarded as the predicted CI areas in order to produce the confusion matrices. Among the four CI detection models, the deterministic RF model showed the highest overall accuracy (94.27 %) and kappa coefficient (89.28 %). The better performance of deterministic RF than DT (overall accuracy of 92.67 % and kappa coefficient of 84.84 %)

is possibly attributed to randomization strategies of deterministic RF such as bootstrap aggregating and randomized node



optimization which can reduce variance and overfitting in building decision trees. The overall accuracy and kappa coefficient value of the probabilistic RF model (94.27 and 89.98 %, respectively) were similar to those of the deterministic RF model because the same randomization strategies were used in the two models. The LR model showed inferior performance in terms of overall accuracy and kappa coefficient (90.63 % and 80.7 2%, respectively) possibly due to the limited capability of the

model to handle the non-linear behavior of the data (Tu, 1996) (Table 6). However, these accuracies using the test dataset could not be generalized to predict real CI cases.

The relative importance of the interest fields that were used for CI detection identified by DT and the deterministic RF are shown in Figs. 3 and 4, respectively. The 11.2 μm $T_B$, representing cloud top temperature, was identified as the most contributing interest field for the discrimination of the two classes (i.e., CI and non-CI) in both DT and deterministic RF

models. This corresponds well with the finding of Mecikalski et al. (2015), which used 25 satellite-based and numerical weather prediction (NWP)-based interest fields to predict CI and identified cloud top temperature as the most contributing satellite-derived variable. The next contributing variables in both DT and deterministic RF models were 13.3-11.2 μm $T_B$ and the trispectral difference time trend (Fig. 3 and 4). The interest field of 13.3-11.2 μm $T_B$ is closely related with cloud top height, which increases as cloud objects evolve into convective ones. Meanwhile, the trispectral difference time trend,

representing the temporal variation of the cloud-top glaciation, increases during the growth of convective clouds. The interest field of 6.2-7.3 μm $T_B$ time trend, the only variable showing a p-value greater than 0.05, was defined as the least contributing interest field in the DT-based CI detection model and the second least one in the deterministic RF model. The relative importance of the interest fields for the LR-based CI detection model can be evaluated using the values of the Exp(b) that are listed in Table 8. The variable of trispectral difference, (8.6-11.2 μm)-(11.2-12.4 μm), has the highest value of the Exp(b),

2.190, which means the probability that a pixel is an actual CI increases by 2.19 times per 1 K increase in the trispectral difference. The 6.2-11.2 μm time trend (Exp(b) = 0.348) was also identified as a contributing variable in the LR model, which decreases the probability that a pixel is an actual CI by 65.2 % per 1 K decrease in the 6.-11.2 μm time trend. These two important variables in the LR model, however, were not ranked high on the variable importance determined by the DT and RF models. The 6.2-11.2 μm time trend was even identified as the least contributing variable in the RF model.

As convective clouds grow vertically in the troposphere (Jorgensen et al., 1989; Trier et al., 2004; Rosenfeld et al., 2008; Sieglaff et al., 2011), time trend variables that represent the vertical growth of clouds might be useful to detect CI. However, the time trend variables among the interest fields here resulted in relatively lower contribution in both DT and RF models. A simple overlap method using two temporally consecutive images was used to determine the temporal change of clouds, instead of AMV that has been widely used to track clouds, because AMV is not yet available for Himawari-8. This may result in

somewhat inaccurate estimation of the vertical growth of clouds. Thus, incorporation of AMV from Himawari-8 in the models may improve the performance of CI detection.





## 4.2 Validation of three CI cases with ground radar and lightning data

The four CI detection models were applied to the three cases of CI events over the Korean Peninsula and validated using two types of reference datasets (i.e., weather radar and lightning observations). Tables 9 and 10 show the validation metrics of the models based on each reference dataset. Fig. 5-8 show CI areas for the case of CI events on 12 June 2015 predicted using the

data collected 0–50 min before CI occurrence by the DT, deterministic RF, probabilistic RF, and LR models, overlaid on a 1.5 km CAPPI image from the case day. For this case day, all models show higher POD, OA, and CSI, but far lower FAR in the validation using the weather radar observations than the lightning. The LR model shows the best performance compared to the other models based on both the radar and lightning observations, producing the highest POD, OA (the second highest in the lightning-based validation), and CSI, and the lowest FAR. Meanwhile, the DT model showed the highest FAR, which would

be caused by the well-known overfitting problem of the algorithm (Pal and Mather, 2003).
Fig. 9-12 show CI areas predicted by the four models for the case of CI events on 1 August 2015, overlaid on a 1.5 km CAPPI image. CI objects on this case day were concentrated on Seoul and Won-san Bay (point locations in the figures), which contributed to lower FAR than other case days. Based on the weather radar observations, the DT model showed the best performance in terms of the POD, OA, and CSI. However, the FAR of the DT model was the highest possibly due to the

overfitting problem. The LR model showed the lowest FAR, but it was not able to detect CI objects around Won-san Bay. Although the periphery of Won-san Bay showed radar echoes above 35 dBZ, CI objects in the region were not included in the validation dataset due to the limited effective radius of the radar. In the validation using lightning observations, all models produced POD of 100 %, except for the probabilistic RF model, all models produced high POD because locations of lightning greatly coincided with those of CI areas and thus large number of hits was produced. However, the FAR produced using the

lightning observations is higher than that using the weather radar. Based on the comparison between the lightning and radar observations, it appears that lightning did not occur for small CI objects.
Fig. 13-16 show CI areas for the case of CI events on 7 August 2015 predicted by the DT, deterministic RF, probabilistic RF, and LR models, respectively, overlaid on a 1.5 km CAPPI image on the day. All models showed better performance in terms of FAR, OA, and CSI based on the weather radar observations than the lightning observations. This is because the number of

CI objects detected by the radar is smaller than that by the lightning observations due to the limited effective radius, similar to the case on 12 June. The DT and deterministic RF models detected CI areas around the northwestern Korean peninsula. Such predicted CI areas might be correct despite the lack of lightning observations, but the weather radar data from China Meteorological Administration (CMA) is not available over the region and hence we were not able to confirm whether the CI objects were correctly identified.

Fig. 13-16 show CI areas for the case of CI events on 7 August 2015 predicted by the DT, deterministic RF, probabilistic RF, and LR models, respectively, overlaid on a 1.5 km CAPPI image on the day. All models showed better performance in terms of FAR, OA, and CSI based on the weather radar observations than the lightning observations. This is because the number of CI objects detected by the radar is smaller than that by the lightning observations due to the limited effective radius, similar to





the case on 12 June. The DT and deterministic RF models detected CI areas around the northwestern Korean peninsula. Such predicted CI areas might be correct despite the lack of lightning observations, but the weather radar data from China Meteorological Administration (CMA) is not available over the region and hence we were not able to confirm whether the CI objects were correctly identified.

The CI detection models developed in this study showed different performances by CI case and the reference dataset used for validation (c.f., Tables 9 and 10). Overall, the DT model produced higher POD than the other models regardless of the validation data. However, it tends to produce high FAR, yielding widely distributed CI in error. The LR model in general produced higher OA and CSI, and lower FAR. However, the POD of the LR model was low due to a large number of missed CI objects compared to the other models. The two validation datasets, i.e., the weather radar and lightning observations,
influenced the assessment of model performance. Since the weather radar sites are located inland, convective clouds over the ocean were out of the detection radius and less likely to be detected. Meanwhile, lightning observations can even detect CI objects over the distant sea, but it is hard to identify the exact location of lightning in clustered-clouds. These limitations in each verification dataset provide the uncertainty in estimating the actual forecast skill of the CI detection models.

     Due to the similar number of hits from the four models, the lead time of all four models was around 32 to 35 min. This indicates
that CI over the Korean Peninsula can be forecasted using the Himawari-8 AHI images with a usable lead time of 30 to 40 min, which is reasonably comparable with the lead time for CI detection (~30–45 min) in the literature (Han et al., 2015; Mecikalski et al., 2015). AHI images 50 min before CI occurrence were used to detect CI in this study. If the AHI data collected a few hours before CI occurrence were used in the development of CI detection models, a longer lead time could possibly be achieved (refer to the supplementary material as an example).

A limitation of this research is in that visible channels, which are critical in identifying cloud tops (Melcikalski et al., 2010), were not used. This was because many of the CI cases over the Korean Peninsula used in this study occurred at night time. The visible channels of Himawari-8 AHI have spatial resolutions of 0.5–1 km which can be used to improve the performance of CI detection models. The use of training samples and validation cases only from summer was another limitation. As the development of convective systems have a clear seasonality (Melcikaski et al., 2010), training samples and validation cases
over different seasons should be incorporated to develop robust CI detection models.

## 5 Conclusions

CI detection models for Himawari-8 AHI data over the Korean Peninsula in East Asia were developed based on DT, RF, and LR. An accuracy assessment of the developed models was conducted using weather radar and lightning observations. The interest field of 11.2 μm $T_B$, representing cloud top temperature, was identified as the most contributing variable in the
deterministic models and the probabilistic RF model. In the LR model, the trispectral difference, i.e., (8.6-11.2 μm)-(11.2-12.4 μm), was identified as the most important one. The developed CI detection models showed varied performance in terms of POD, FAR, OA, and CSI by CI cases and validation datasets. Nevertheless, the DT model produced generally higher POD



than the other models, while the LR model showed higher OA and CSI, and lower FAR. The averaged lead time of the CI detection models was calculated between 32 and 35 min, which means that a 30-min forecast of CI over the Korean Peninsula during summer is possible when using Himawari-8 AHI data.

The overlying method could produce a potential error when calculating the time trend interest fields that represent the vertical

growth of convective clouds. AMV might mitigate the error and enhance the availability of time trend variables. Future research includes (1) improving CI detection algorithms using visible reflectance with 0.5 km resolution, (2) increasing training samples and validation cases to reflect diverse convective environments with different seasons, and (3) expanding the period of the Himawari-8 AHI data 2 hours before CI occurrence.

Acknowledgements. This work was supported by the "Development of Geostationary Meteorological Satellite Ground Segment (NMSC-2014-01)" program funded by the National Meteorological Satellite Centre (NMSC) of the Korea Meteorological Administration (KMA).

**Table 1**. Characteristics of the spectral channels of Himawari-8 AHI

| No. Band | Central wavelength (µm) | Bandwidth (µm) | Spatial resolution (km) |
|---|---|---|---|
| 1 | 0.455 | 0.05 | 1 |
| 2 | 0.510 | 0.02 | 1 |
| 3 | 0.645 | 0.03 | 0.5 |
| 4 | 0.86 | 0.02 | 1 |
| 5 | 1.61 | 0.02 | 2 |
| 6 | 2.26 | 0.02 | 2 |
| 7 | 3.85 | 0.22 | 2 |
| 8 | 6.25 | 0.37 | 2 |
| 9 | 6.95 | 0.12 | 2 |
| 10 | 7.35 | 0.17 | 2 |
| 11 | 8.60 | 0.32 | 2 |
| 12 | 9.63 | 0.18 | 2 |
| 13 | 10.45 | 0.30 | 2 |
| 14 | 11.20 | 0.20 | 2 |
| 15 | 12.35 | 0.30 | 2 |
| 16 | 13.30 | 0.20 | 2 |





**Table 2**. Convective initiation (CI) cases used to develop and validate the deterministic and probabilistic CI detection models

| ID | Date | Time (hh:mm, UTC) | Usage |
|----|------|-------------------|-------|
| 1 | 13 June 2015 | 09:40 | |
| 2 | 13 June 2015 | 11:10 | |
| 3 | 16 June 2015 | 14:30 | |
| 4 | 17 June 2015 | 10:30 | Training dataset |
| 5 | 4 July 2015 | 06:30 | |
| 6 | 25 July 2015 | 00:10 | |
| 7 | 16 August 2015 | 10:30 | |
| 8 | 12 June 2015 | 14:30 | |
| 9 | 1 August 2015 | 19:10 | Validation dataset |
| 10 | 7 August 2015 | 08:00 | |

**Table 3**. Summary of the interest fields to develop convective initiation (CI) detection models used in this study (Walker and Mecikalski, 2011)

| ID | Interest field | Contribution |
|----|----------------|--------------|
| 1 | 11.2 μm $T_B$ | Cloud-top temperature assessment |
| 2 | 6.2-11.2 μm | |
| 3 | 6.2-7.3 μm | Cloud-top height relative to tropopause |
| 4 | 13.3-11.2 μm | |
| 5 | 12.3-11.2 μm | |
| 6 | 8.6-11.2 μm | Cloud-top glaciation |
| 7 | 11.2 μm time trend | Cloud-top cooling rate |
| 8 | 6.2-11.2 μm time trend | |
| 9 | 6.2-7.3 μm time trend | Temporal changes in cloud-top height |
| 10 | 12.3-11.2 μm time trend | |
| 11 | (8.6-11.2 μm)-(11.2-12.3 μm) | Cloud-top glaciation |
| 12 | (8.6-11.2 μm)-(11.2-12.3 μm) time trend | Temporal changes in cloud-top glaciation |



**Table 4.** Assessment of the DT model for CI detection using the test data.

| Reference  Classification | CI | Non-CI | Sum | User's accuracy |
|---|---|---|---|---|
| CI | 239 | 21 | 260 | 91.92 % |
| Non-CI | 26 | 355 | 381 | 93.18 % |
| Sum | 265 | 376 | 641 | |
| Producer's accuracy | 90.19 % | 91.41 % | | |
| Overall accuracy | | | 92.67 % | |
| Kappa coefficient | | | 84.84 % | |

**Table 5.** Assessment of the RF model for CI detection using the test data.

| Reference  Classification | CI | Non-CI | Sum | User's accuracy |
|---|---|---|---|---|
| CI | 245 | 11 | 256 | 95.70 % |
| Non-CI | 20 | 365 | 385 | 94.81 % |
| Sum | 265 | 376 | 641 | |
| Producer's accuracy | 92.45 % | 97.07 % | | |
| Overall accuracy | | | 94.27 % | |
| Kappa coefficient | | | 89.98 % | |

**Table 6.** Assessment of the probabilistic RF model for CI detection using the test dataset. CI probabilities above 50% considered as a CI.

| Reference  Classification | CI | Non-CI | Sum | User's accuracy |
|---|---|---|---|---|
| CI | 242 | 13 | 255 | 94.90 % |
| Non-CI | 23 | 363 | 386 | 94.04 % |
| Sum | 265 | 376 | 641 | |
| Producer's accuracy | 91.32 % | 96.54 % | | |
| Overall accuracy | | | 94.38 % | |
| Kappa coefficient | | | 88.36 % | |



**Table 7.** Assessment of the LR model for CI detection using the test dataset. CI probabilities above 50% considered as a CI.

| Reference<br>Classification | CI | Non-CI | Sum | User's accuracy |
|---|---|---|---|---|
| CI | 237 | 33 | 270 | 82.78 % |
| Non-CI | 27 | 343 | 370 | 92.70 % |
| Sum | 264 | 376 | 640 | |
| Producer's accuracy | 89.77 % | 91.22 % | | |
| Overall accuracy | | | 90.63 % | |
| Kappa coefficient | | | 80.72 % | |

**Table 8.** Exp (b) values from the logistic regression (LR) model, which are odds ratios derived by SPSS at the significance level 95 %.

| Interest field | Exp (b) |
|---|---|
| (8.6-11.2 μm)-(11.2-12.4 μm) | 2.190 |
| 12.4-11.2 μm time trend | 1.456 |
| 6.2-11.2 μm | 1.146 |
| 13.3-11.2 μm | 1.059 |
| 6.2-7.3 μm time trend | 0.979 |
| (8.6-11.2 μm)-(11.2-12.4 μm) time trend | 0.979 |
| 6.2-7.3 μm | 0.923 |
| 12.4-11.2 μm | 0.802 |
| 11.2 μm | 0.670 |
| 11.2 μm time trend | 0.585 |
| 8.6-11.2 μm | 0.483 |
| 6.2-11.2 μm time trend | 0.348 |



**Table 9**. Validation metrics and averaged lead times based on radar CAPPI data for the deterministic DT and RF, probabilistic RF, and LR models.

|  |  | 12 June 2015 | 1 August 2015 | 7 August 2015 | Overall |
|---|---|---|---|---|---|
| **DT** | POD (%) | 89.10 | 87.37 | 85.1 | 87.19 |
|  | FAR (%) | 57.74 | 18.2 | 34.42 | 36.78 |
|  | OA (%) | 90.59 | 90.62 | 83.87 | 88.36 |
|  | CSI (%) | 40.17 | 73.17 | 58.82 | 57.39 |
|  | Lead time (min) | 35 | 35 | 34.58 | 34.86 |
| **Deterministic RF** | POD (%) | 87.12 | 67.16 | 85.92 | 80.07 |
|  | FAR (%) | 41.72 | 1.1 | 27.95 | 23.59 |
|  | OA (%) | 88.67 | 90.58 | 86.88 | 88.71 |
|  | CSI (%) | 53.65 | 66.7 | 64.44 | 61.58 |
|  | Lead time (min) | 34.8 | 35.26 | 34.31 | 34.73 |
| **Probabilistic RF** | POD (%) | 84.84 | 75.22 | 86.61 | 82.23 |
|  | FAR (%) | 47.16 | 3.5 | 28.9 | 26.53 |
|  | OA (%) | 84.65 | 82.63 | 86.2 | 84.49 |
|  | CSI (%) | 48.27 | 73.21 | 64 | 61.85 |
|  | Lead time (min) | 34.23 | 33.23 | 34.79 | 34.64 |
| **LR** | POD (%) | 90.19 | 59.37 | 69.56 | 73.04 |
|  | FAR (%) | 32.84 | 1 | 26.15 | 15 |
|  | OA (%) | 92.68 | 81.46 | 84.1 | 87.43 |
|  | CSI (%) | 62.58 | 59 | 55.81 | 61.86 |
|  | Lead time (min) | 33.6 | 34.14 | 32.5 | 33.81 |



**Table 10**. Validation metrics and averaged lead times based on lightning data for the deterministic DT and RF, probabilistic RF, and LR models.

| | | 12 June 2015 | 1 August 2015 | 7 August 2015 | Overall |
|---|---|---|---|---|---|
| **DT** | POD (%) | 81.48 | 98.7 | 87.5 | 89.66 |
| | FAR (%) | 84.61 | 44.32 | 46.83 | 58.59 |
| | OA (%) | 44.52 | 84.9 | 77.48 | 69..30 |
| | CSI (%) | 14.86 | 55.67 | 49.41 | 39.98 |
| | Lead time (min) | 35 | 35 | 35 | 34.39 |
| **Deterministic RF** | POD (%) | 77.41 | 98.9 | 78.74 | 85.29 |
| | FAR (%) | 73.03 | 29.87 | 35.42 | 46.11 |
| | OA (%) | 48.62 | 91.12 | 81.32 | 74.02 |
| | CSI (%) | 25 | 70.12 | 54.85 | 49.94 |
| | Lead time (min) | 33.09 | 35 | 32.68 | 34.55 |
| **Probabilistic RF** | POD (%) | 80.35 | 96.29 | 79.22 | 81.29 |
| | FAR (%) | 74.28 | 38.09 | 39.90 | 50.76 |
| | OA (%) | 45.62 | 88.11 | 78.63 | 70.79 |
| | CSI (%) | 24.19 | 60.46 | 51.91 | 45.52 |
| | Lead time (min) | 33.18 | 33.4 | 34.26 | 33.61 |
| **LR** | POD (%) | 85.18 | 98.1 | 67.36 | 84.18 |
| | FAR (%) | 72.61 | 14.28 | 32.63 | 39.84 |
| | OA (%) | 46.64 | 96.66 | 81.31 | 74.87 |
| | CSI (%) | 26.13 | 85.2 | 50.78 | 54.21 |
| | Lead time (min) | 34.2 | 33.4 | 34.26 | 33.62 |



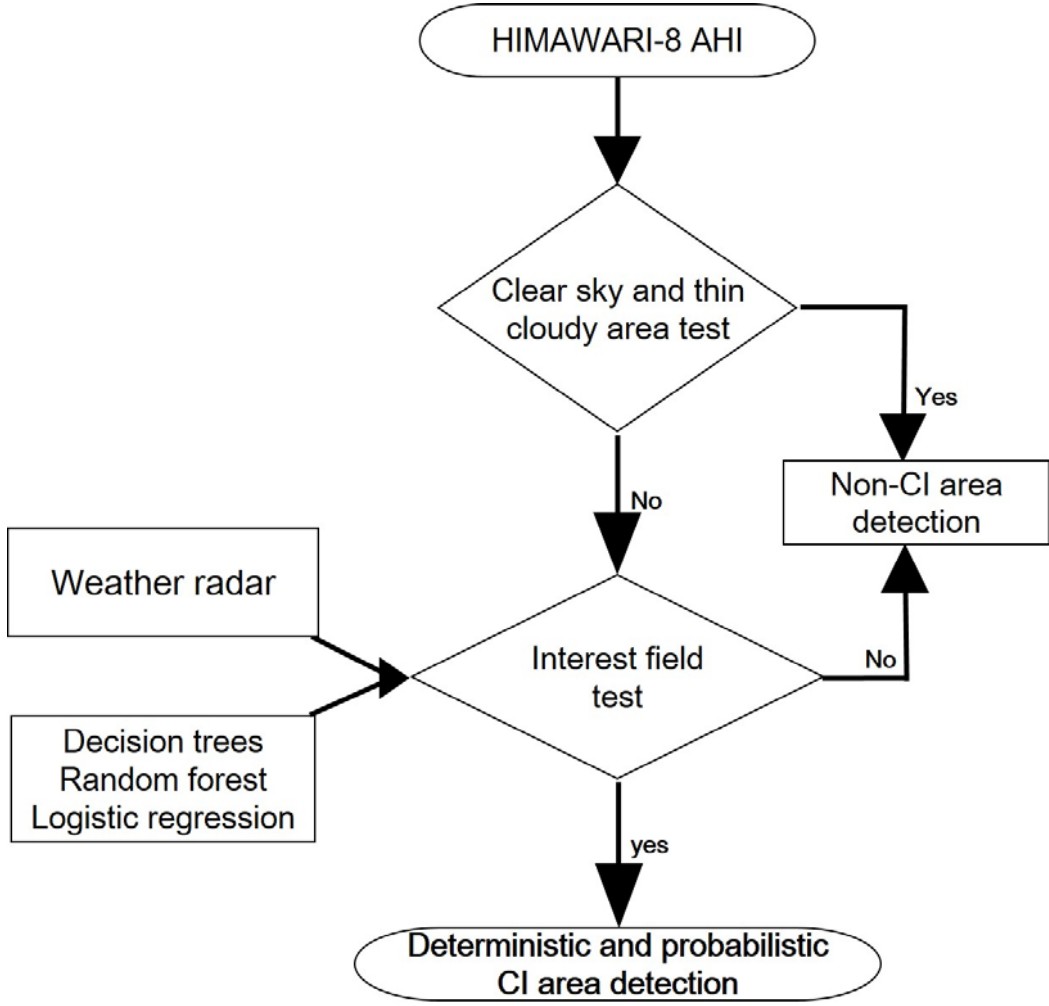

**Figure 1.** Processing flow of deterministic and probabilistic CI detection based on machine learning and statistical methods.





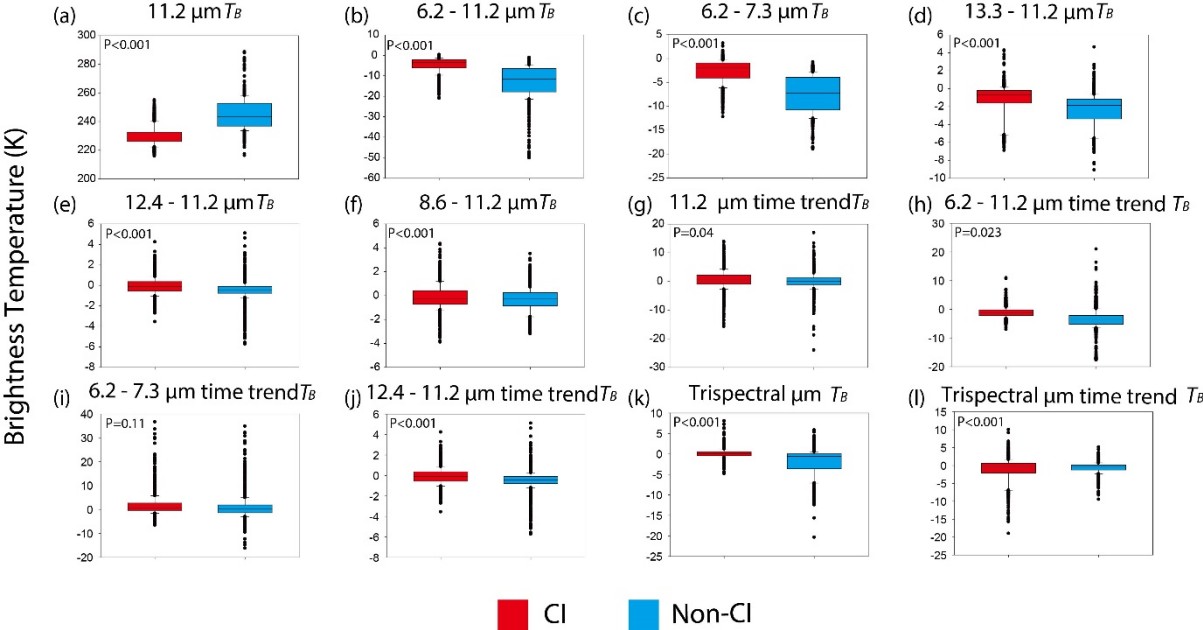

**Figure 2.** Box plots of 12 input variables (i.e., interest fields) generated based on the reference data used for CI detection models: (a) 11.2 μm $T_b$, (b) 6.2 − 11.2 μm $T_b$, (c) 6.2 − 7.3 μm $T_b$, (d) 13.3 − 11.2 μm $T_b$, (e) 12.4 − 11.2 μm $T_b$, (f) 8.6 − 11.2 μm $T_b$, (g) 11.2 μm time trend $T_b$, (h) 6.2 − 11.2 μm time trend $T_b$, (i) 6.2 − 7.3 μm time trend $T_b$, (j) 12.4 − 11.2 μm time trend $T_b$, (k) Tri-spectral μm $T_b$, (l) Tri-spectral μm time trend $T_b$.





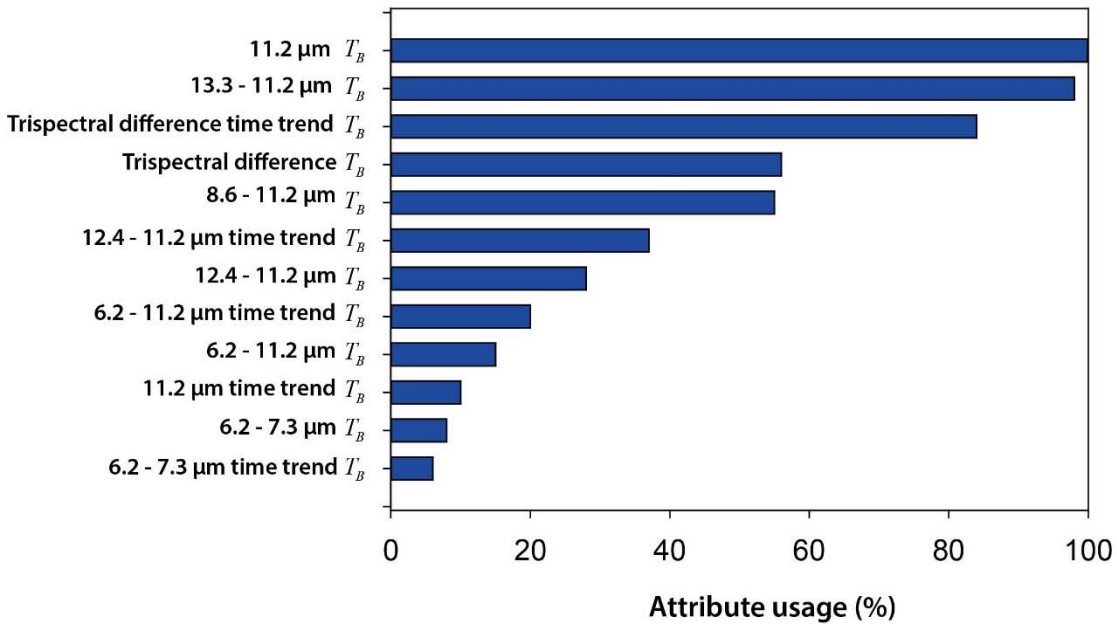

**Figure 3.** Attribute usage information in percentage by interest field produced in the DT model.



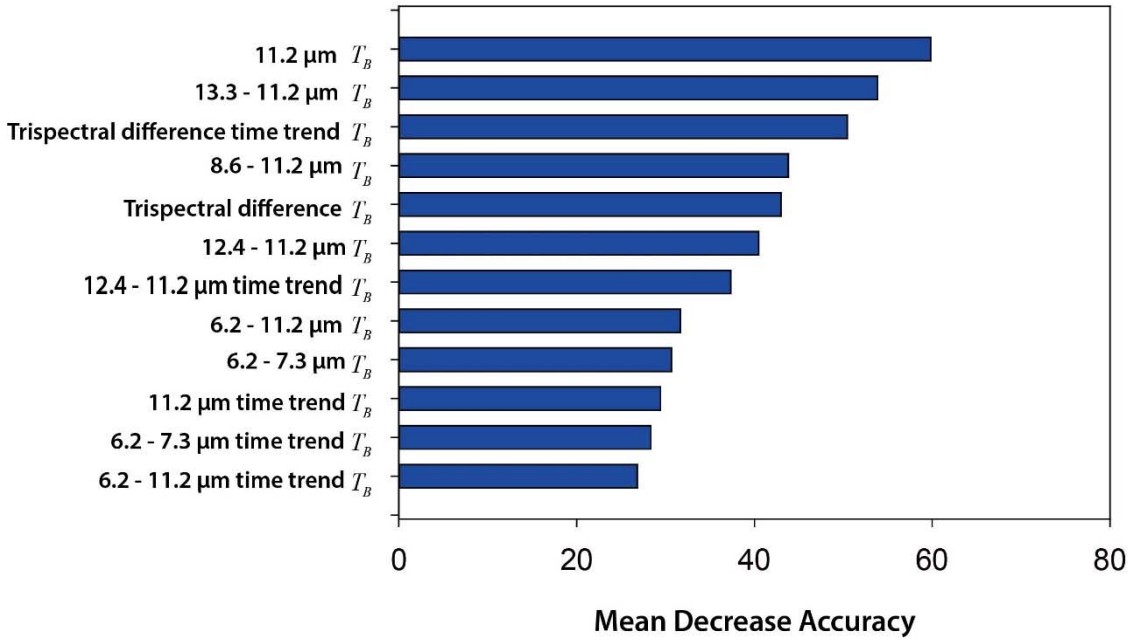

**Figure 4.** Mean Decrease Accuracy (MDA) in percentage by interest field produced in the RF model. MDA is calculated using out-of-bag (OOB) data when an interest field was randomly permuted. The higher the MDA of an interest field, the more the field contributes to identify CI.



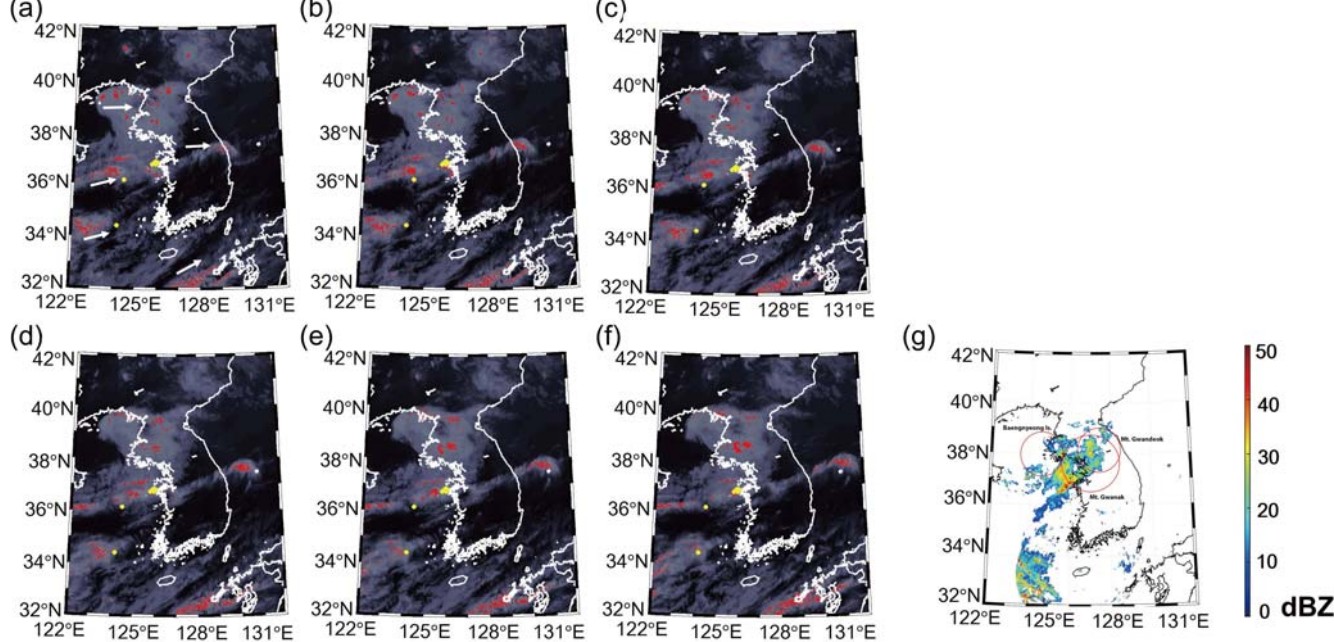

**Figure 5.** Deterministic CI detection map derived by DT on 12 June 2015 13:40 – 14:30 (UTC): (a) 50 min, (b) 40 min, (c) 30 min, (d), 20 min, (e) 10 min, and (f) 0 min before CI occurrence. (g) is 1.5 km radar CAPPI image at 14:30 (UTC). While predicted CI is in red, the locations of lightning occurrences at 14:30 (UTC) are presented in yellow dots. Red circles in (g) mean the effective radius of each radar site. The averaged direction of cloud objects is shown with white arrows in (a).





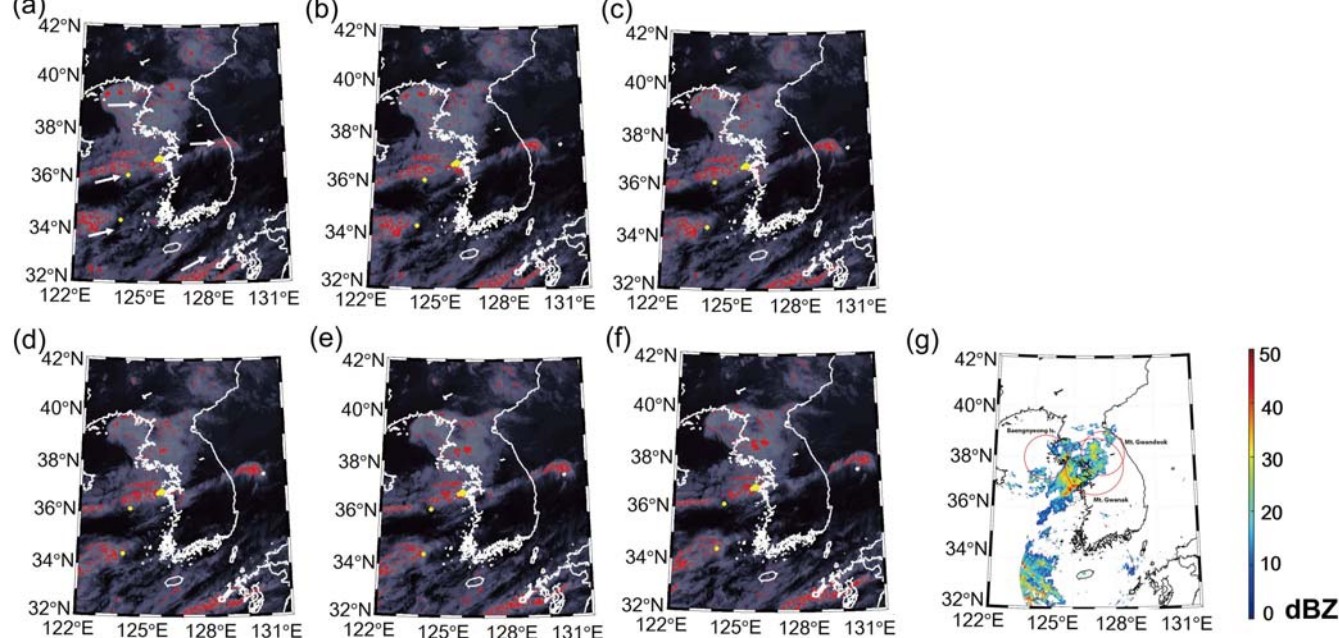

**Figure 6.** Deterministic CI detection map derived by deterministic RF on 12 June 2015 13:40 – 14:30 (UTC): (a) 50 min, (b) 40 min, (c) 30 min, (d), 20 min, (e) 10 min, and (f) 0 min before CI occurrence. (g) is 1.5 km radar CAPPI image at 14:30 (UTC). While predicted CI is in red, the locations of lightning occurrences at 14:30 (UTC) are presented in yellow dots. Red circles in (g) mean the effective radius of each radar site. The averaged direction of cloud objects is shown with white arrows in (a).



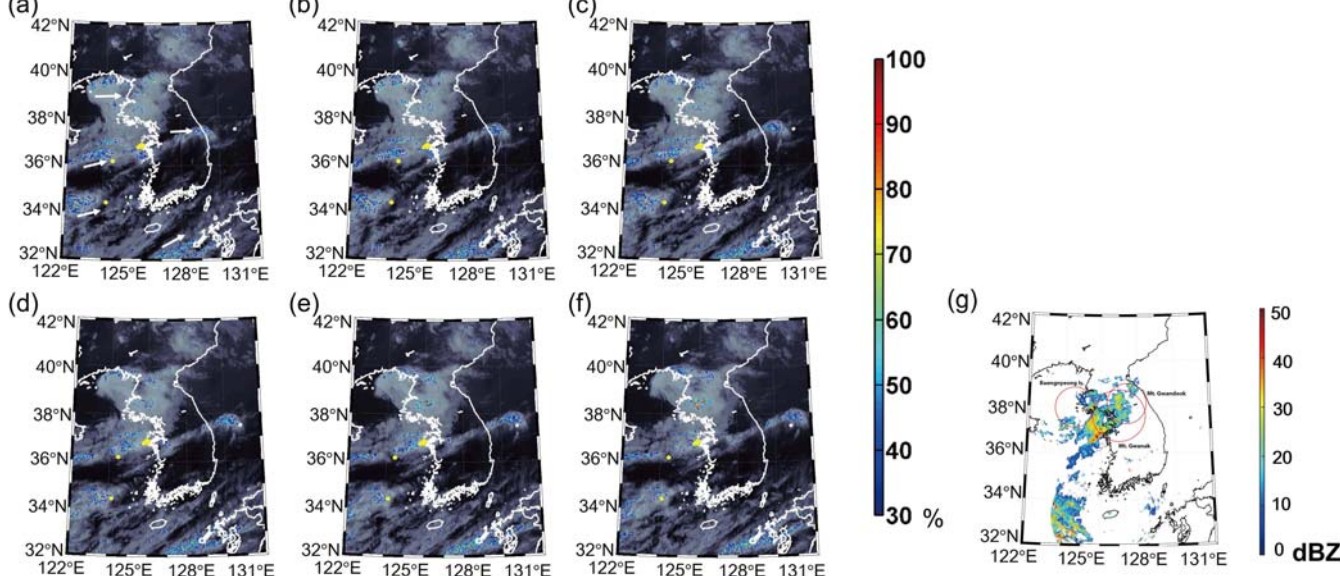

**Figure 7.** Probabilistic CI detection map derived by RF on 12 June 2015 13:40 – 14:30 (UTC): (a) 50 min, (b) 40 min, (c) 30 min, (d), 20 min, (e) 10 min, and (f) 0 min before CI occurrence. (g) is 1.5 km radar CAPPI image at 14:30 (UTC). While predicted CI is in red, the locations of lightning occurrences at 14:30 (UTC) are presented in yellow dots. Red circles in (g) mean the effective radius of each radar site. The averaged direction of cloud objects is shown with white arrows in (a).



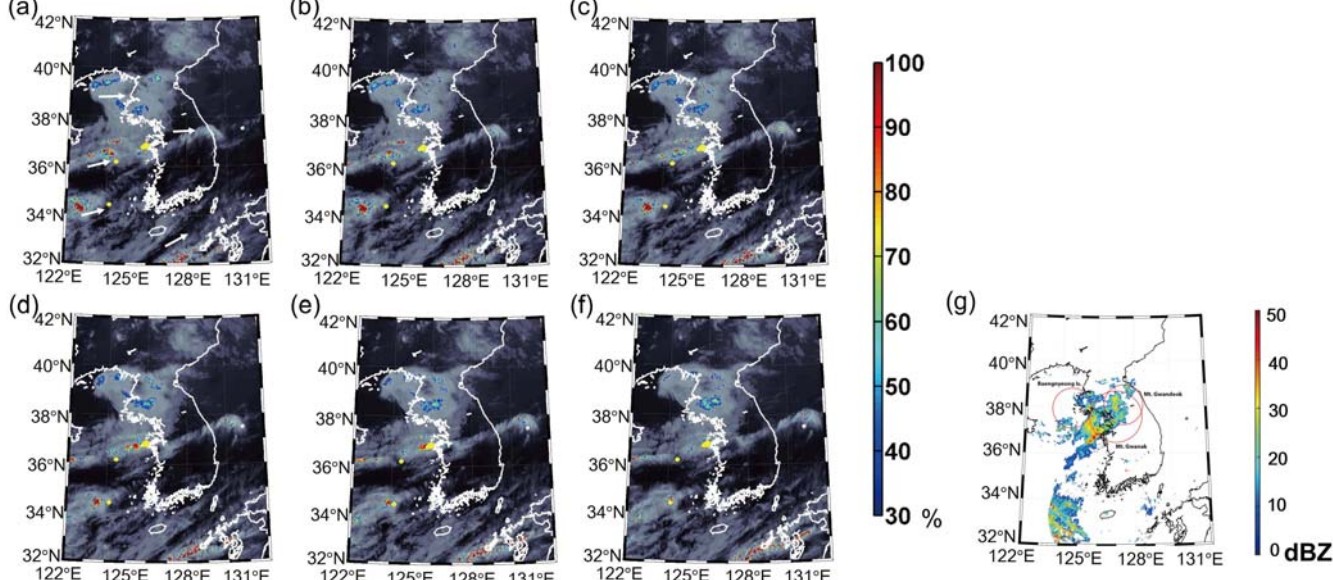

**Figure 8.** Probabilistic CI detection map derived by LR on 12 June 2015 13:40 – 14:30 (UTC): (a) 50 min, (b) 40 min, (c) 30 min, (d), 20 min, (e) 10 min, and (f) 0 min before CI occurrence. (g) is 1.5 km radar CAPPI image at 14:30 (UTC). While predicted CI is in red, the locations of lightning occurrences at 14:30 (UTC) are presented in yellow dots. Red circles in (g) mean the effective radius of each radar site. The averaged direction of cloud objects is shown with white arrows in (a).



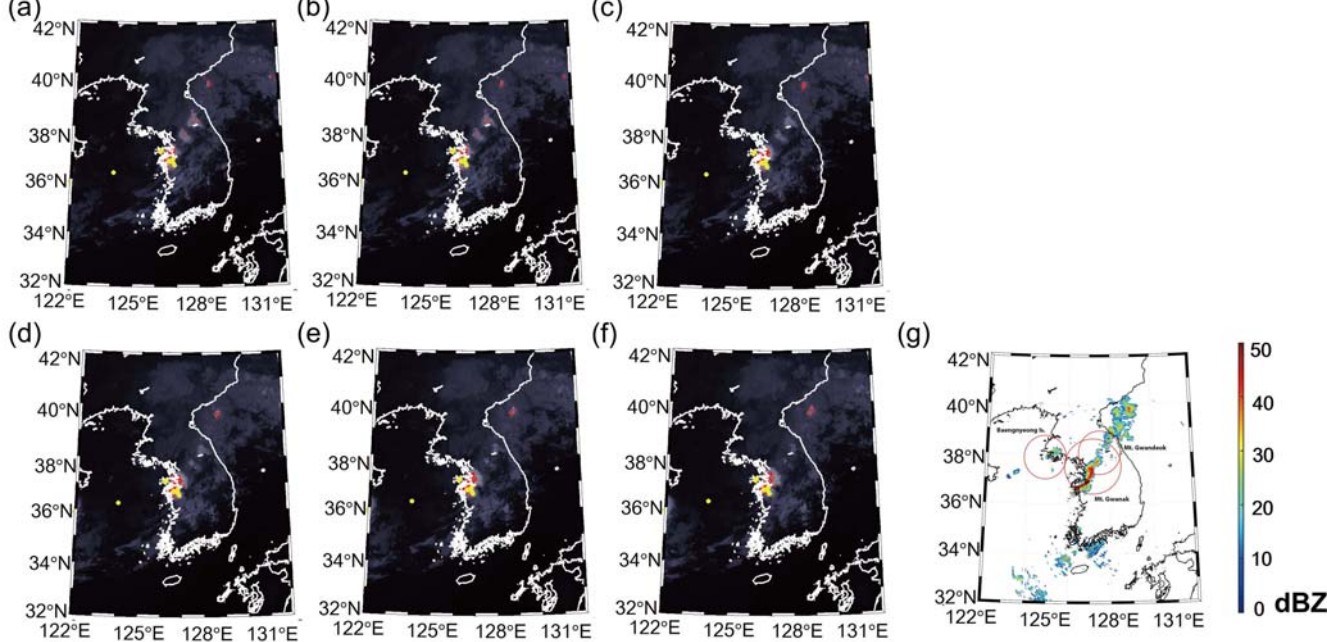

**Figure 9.** Deterministic CI detection map derived by DT on 1 August 2015 1810 - 1910 (UTC): (a) 50 min, (b) 40 min, (c) 30 min, (d), 20 min, (e) 10 min, and (f) 0 min before CI occurrence. (g) is 1.5 km radar CAPPI image at 19:10 (UTC). While predicted CI is in red, the locations of lightning occurrences at 19:10 (UTC) are presented in yellow dots. Red circles in (g) mean the effective radius of each radar site. The averaged direction of cloud objects is shown with white arrows in (a).



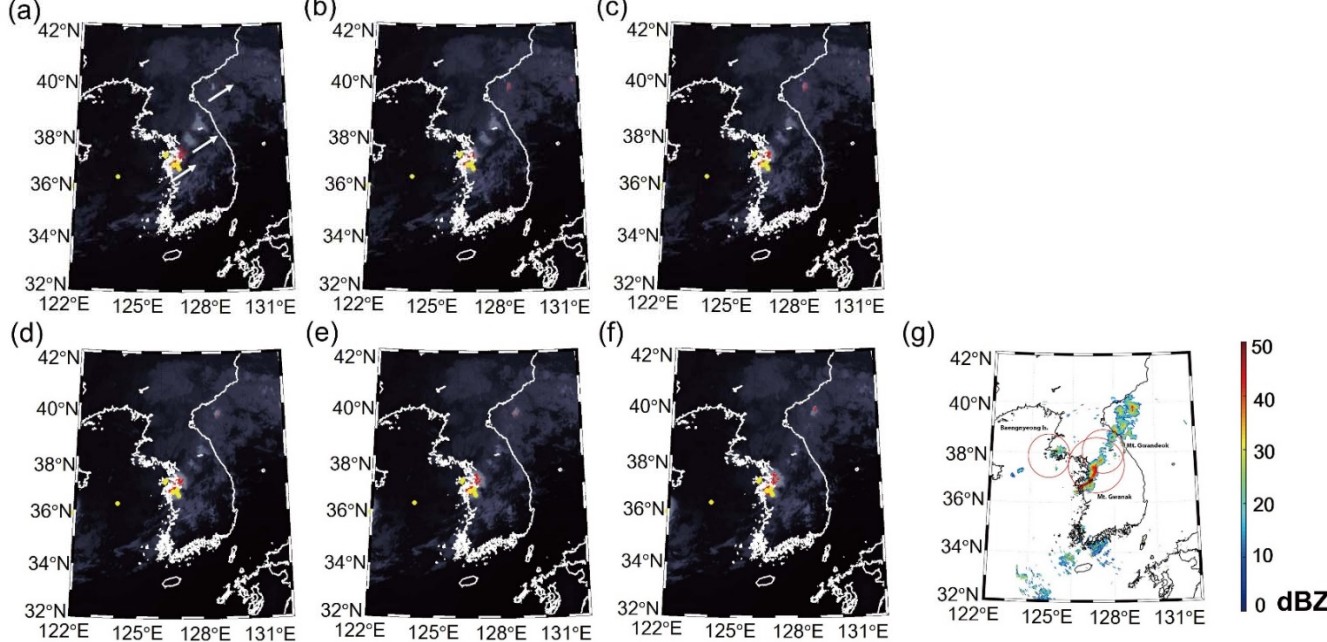

**Figure 10.** Deterministic CI detection map derived by RF on 1 August 2015 1810 - 1910 (UTC): (a) 50 min, (b) 40 min, (c) 30 min, (d), 20 min, (e) 10 min, (f) 0 min before CI occurrence. (g) is 1.5 km radar CAPPI image at 19:10 (UTC). While predicted CI is in red, the locations of lightning occurrences at 19:10 (UTC) are presented in yellow dots. Red circles in (g) mean the effective radius of each radar site. The averaged direction of cloud objects is shown with white arrows in (a).



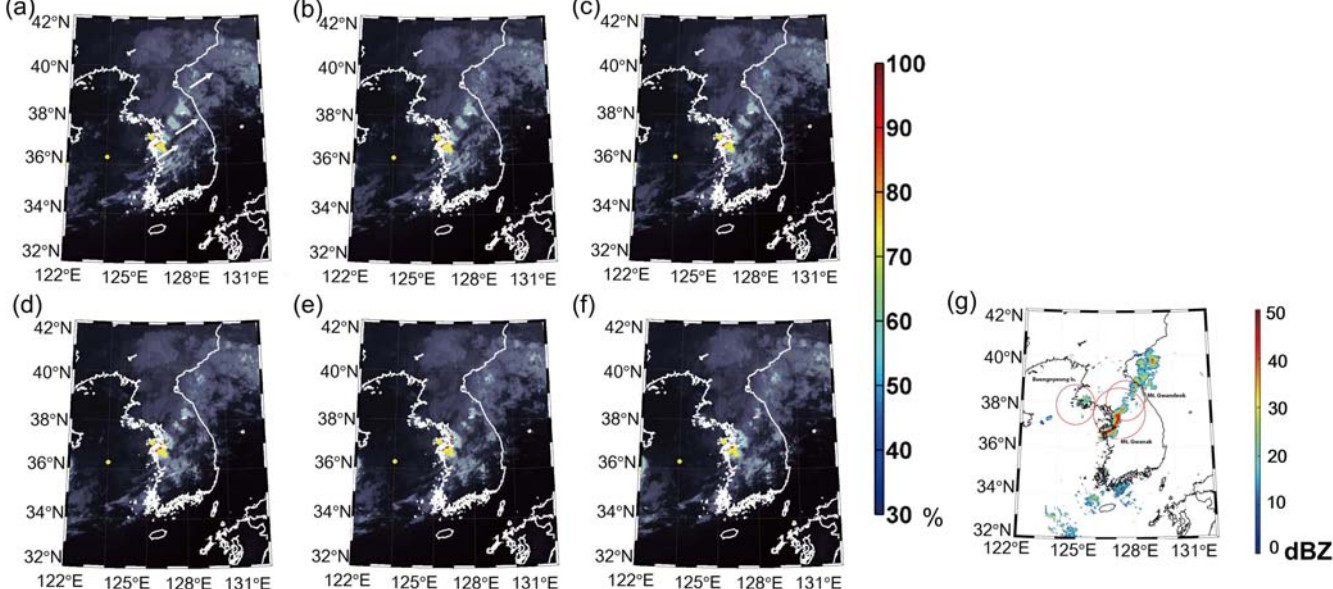

**Figure 11.** Probabilistic CI detection map derived by RF on 1 August 2015 1810 - 1910 (UTC): (a) 50 min, (b) 40 min, (c) 30 min, (d), 20 min, (e) 10 min, (f) 0 min before CI occurrence. (g) is 1.5 km radar CAPPI image at 19:10 (UTC). While predicted CI is in red, the locations of lightning occurrences at 19:10 (UTC) are presented in yellow dots. Red circles in (g) mean the effective radius of each radar site. The averaged direction of cloud objects is shown with white arrows in (a).



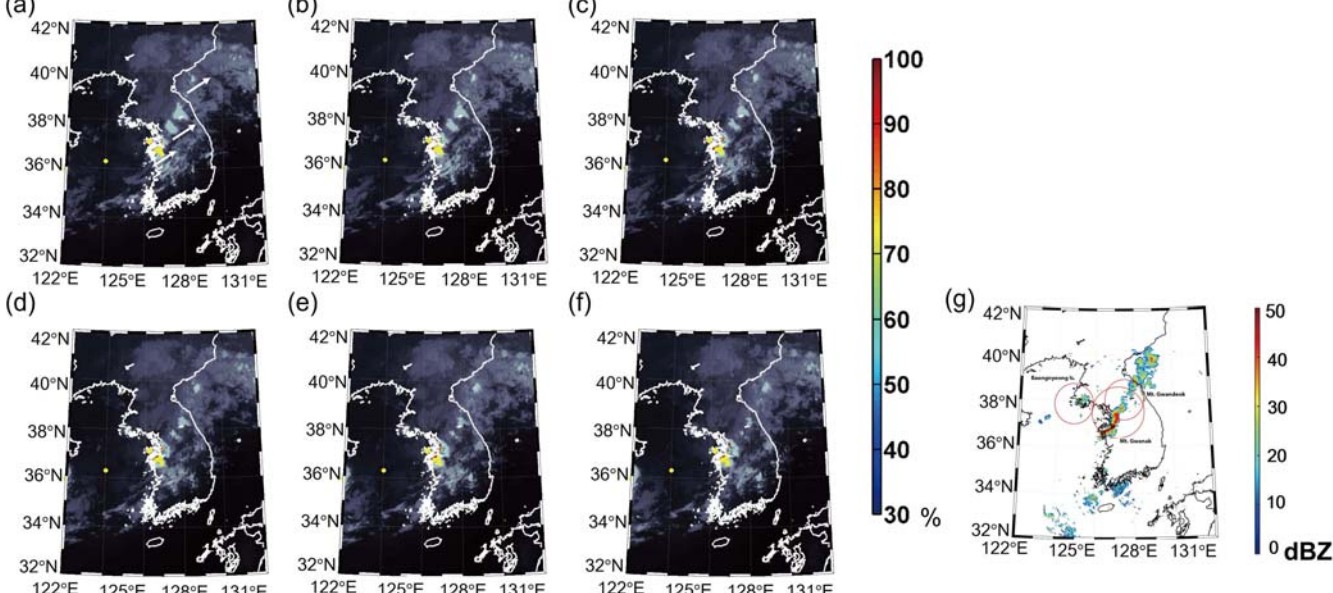

**Figure 12.** Probabilistic CI detection map derived by LR on 1 August 2015 1810 - 1910 (UTC): (a) 50 min, (b) 40 min, (c) 30 min, (d), 20 min, (e) 10 min, (f) 0 min before CI occurrence. (g) is 1.5 km radar CAPPI image at 19:10 (UTC). While predicted CI is in red, the locations of lightning occurrences at 19:10 (UTC) are presented in yellow dots. Red circles in (g) mean the effective radius of each radar site. The averaged direction of cloud objects is shown with white arrows in (a).




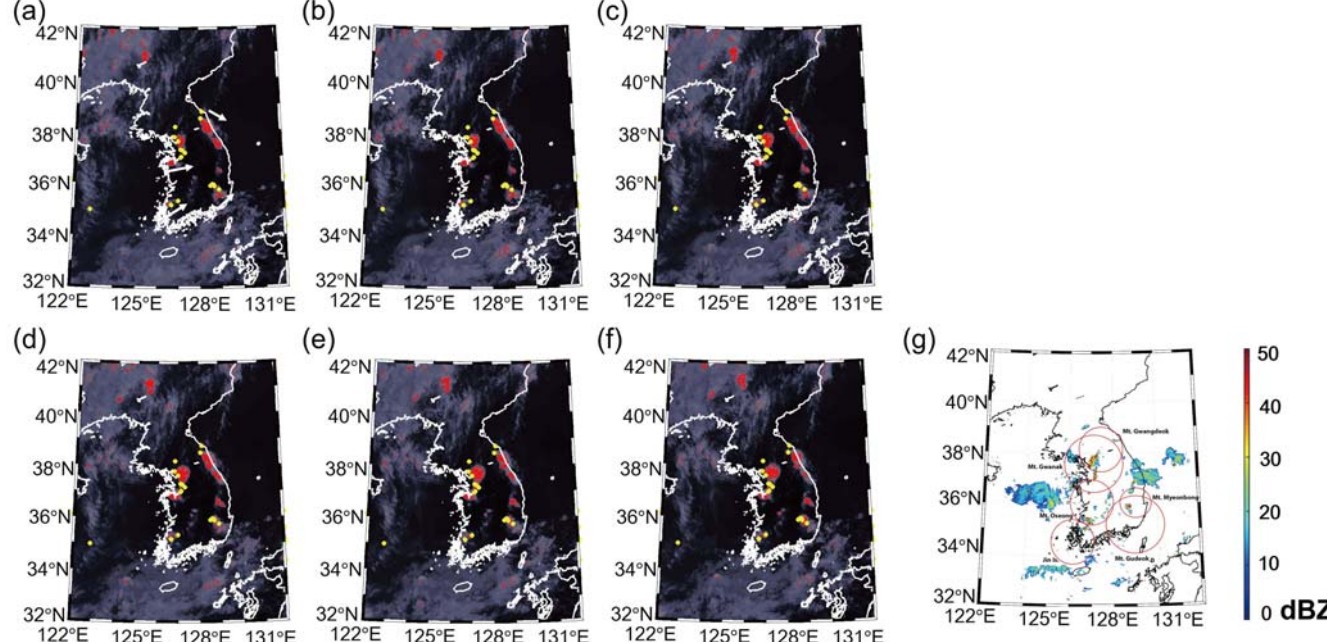

**Figure 13.** Deterministic CI detection map derived by DT on 7 August 2015 0710 - 0800 (UTC): (a) 50 min, (b) 40 min, (c) 30 min, (d), 20 min, (e) 10 min, (f) 0 min before CI occurrence. (g) is 1.5 km radar CAPPI image at 08:00 (UTC). While predicted CI is in red, the locations of lightning occurrences at 08:00 (UTC) are presented in yellow dots. Red circles in (g) mean the effective radius of each radar site. The averaged direction of cloud objects is shown with white arrows in (a).



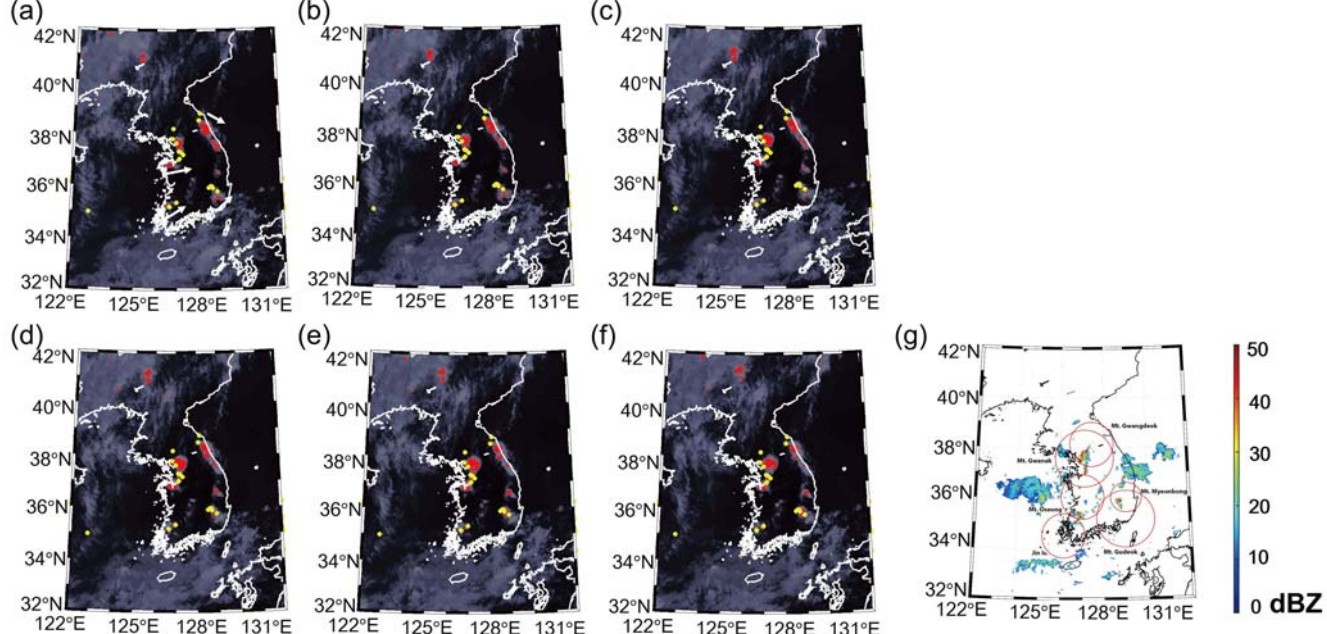

**Figure 14.** Deterministic CI detection map derived by RF on 7 August 2015 0710 - 0800 (UTC): (a) 50 min, (b) 40 min, (c) 30 min, (d), 20 min, (e) 10 min, (f) 0 min before CI occurrence. (g) is 1.5 km radar CAPPI image at 08:00 (UTC). While predicted CI is in red, the locations of lightning occurrences at 08:00 (UTC) are presented in yellow dots. Red circles in (g) mean the effective radius of each radar site. The averaged direction of cloud objects is shown with white arrows in (a).



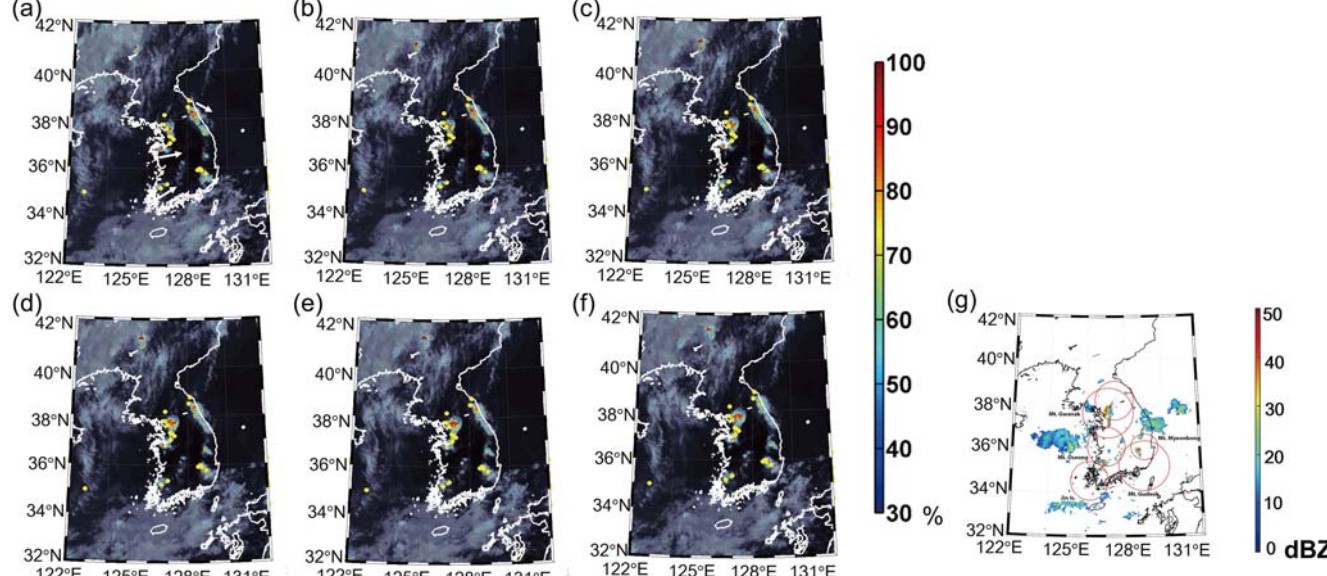

**Figure 15.** Probabilistic CI detection map derived by RF on 7 August 2015 0710 - 0800 (UTC): (a) 50 min, (b) 40 min, (c) 30 min, (d), 20 min, (e) 10 min, and (f) 0 min before CI occurrence. (g) is 1.5 km radar CAPPI image at 08:00 (UTC). While predicted CI is in red, the locations of lightning occurrences at 08:00 (UTC) are presented in yellow dots. Red circles in (g) mean the effective radius of each radar site. The averaged direction of cloud objects is shown with white arrows in (a).



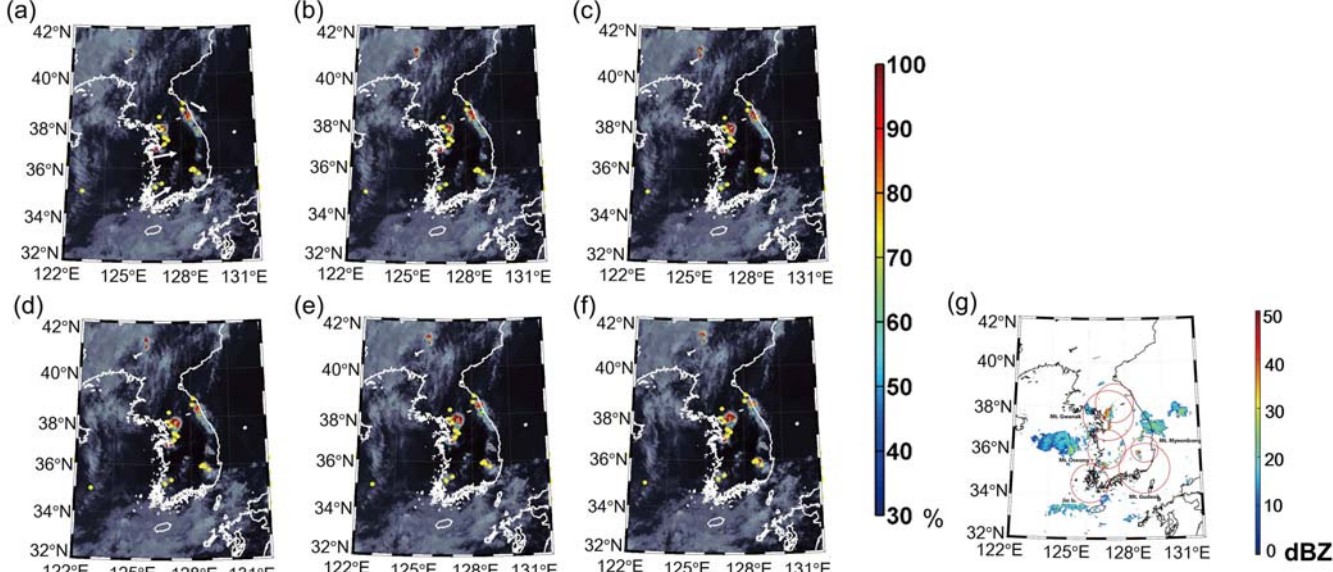

**Figure 16.** Probabilistic CI detection map derived by LR on 7 August 2015 0710 - 0800 (UTC): (a) 50 min, (b) 40 min, (c) 30 min, (d), 20 min, (e) 10 min, and (f) 0 min before CI occurrence. (g) is 1.5 km radar CAPPI image at 08:00 (UTC). While predicted CI is in red, the locations of lightning occurrences at 08:00 (UTC) are presented in yellow dots. Red circles in (g) mean the effective radius of each radar site. The averaged direction of cloud objects is shown with white arrows in (a).



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
