# Peer review of "Detection of deterministic and probabilistic convective initiation using Himawari-8 Advanced Himawari Imager data"

_Atmospheric Measurement Techniques, 2016_

## Referee Comment (RC1) · Anonymous Referee #1 · 16 Dec 2016

Note: I review this same paper in July/August 2016, and as it looks, only a few of my major and minor comments sent back to the authors at that time have been addressed.

This paper presents the implementation of a convective initiation (CI) algorithm that operates on Himawari-8 AHI data, with applications to the Korean Peninsula. Overall, a major concern related to the requirement that journals present NEW, innovative work is that this paper repeats much of the analysis methods in two papers, Mecikalski et al. (2010) and Mecikalski et al. (2015). If this journal is o.k. with the "newness" being that prior methods are developed using new (i.e. Himawari-8) data and not new methods, then the paper is not a bad presentation. But, the paper suffers from considerable grammatical problems. It need to be read by an English speaking person.

In the Mecikalski et al. (2010) paper, several Meteosat Second Generation (MSG) satellite based fields were defined outward of principal component and other information content analysis (that began with an assessment of many possible interest fields) for their value at predicting CI in the coming 1 hour. These are the results as stated in their Table 3. In the Mecikalski et al. (2015) paper, which defines the current GOES-R CI algorithm (albeit the authors have some confusion as to the authors/developers of the GOES-R CI algorithm—see corrections below), the RF and LR machine learning approaches were applied to a reduced set of GOES-specific satellite predictor fields and also to NWP fields. Hence, this paper seemingly combined the Mecikalski et al. (2010) and Mecikalski et al. (2015) studies, with little new information, insights or analysis being done. I therefore again am not sure how appropriate it is to publish such a study that just re-applies already-published ideas, yet the authors have properly cited the relevant prior research and that the results are similar to these prior works.

I reiterate that it may not be appropriate to publish results which are effectively ∼95%+ stated in other papers, based on the Methodology of Section 3, and the results in Tables 9 and 10 are also nearly the same for the RF and LR models as shown in Mecikalski et al. (2015). These authors have not done a complete analysis of all possible AHI datasets with respect to CI, which is in fact an ongoing activity in my research group today, and therefore do not shed new light on the value of all 13 infrared and the visible channels for predicting CI. I am therefore inclined to Reject this paper since it effectively duplicates prior work, while not significantly advancing our understanding.

If the Editor deems it appropriate to publish these results, then...

(1) The authors need to carefully define all acronyms before they are used. This problem begins in the abstract and continues well into the paper. There are numerous spelling and grammatical issues (e.g., Himawari-8 is not capitalized on page 4/top, and the correct acronym is "EUMETSAT" on "EUMESAT" on page 2).

(3) Overall, it is not good to validate a CI algorithm using lightning observations, especially in the algorithm development stages. The authors do note the reason for poorer model performance when lightning data were used, later in the paper, as not all CI events go on to make lightning.

(4) Methodology/Section 3: Please remove the sentence "However, as mentioned, the GOES-R CI algorithm uses simple threshold values associated with the interest fields and the values were determined through many experimental simulations in a subjective way." First, the present GOES-R CI algorithm (Mecikalski et al. 2015) does NOT use "simple threshold values". Second, all prior research was NOT subjective in nature, but rather examined growing cumulus clouds in advance of CI with respect to physical processes (cloud growth rates, updraft size/width, glaciation, cloud altitude, updraft longevity, etc.) as measured/observed by geostationary satellite infrared and visible datasets. Specifically, the Mecikalski et al. (2010) paper using MSG data was very focused on gaining understanding on how the interest fields behaved and subsequently how specific "threshold" values could be set, similar to how the original Mecikalski and Bedka (2006) study was performed. The way this sentence reads is that the prior work was just done without much thought, which was hardly the case.

For more MINOR comments, I again suggest that the paper be reviewed and edited by an English-speaking person prior to acceptance.

---

## Referee Comment (RC2) · Anonymous Referee #2 · 2 Feb 2017

This paper presents studies related to the use and selection of a convective initiation (CI) algorithm for application to Himawari-8 AHI data, specifically collected for the Korean Peninsula.

The paper addresses questions within the scope of AMT although it does not introduce new concepts or ideas. It reaches interesting conclusions in the context of applying the data to the Korean Peninsula and, although the novelty of the paper is minimal, it gives a reasonable description of the issues involved with detecting CI. Publication of such analyses is not unusual for new instrumentation as it assists others and it provides a benchmark in the analysis process. In this case, the the data analysis is limited in the confidence which we can have by the small number of days (in the figures) for which

the results of the training data are applied. If the algorithms are truly to be 'validated over Northeast Asia', we need a better (larger) validation data-set.

The English language in the paper would benefit from the advice of a native english speaker but it is not disastrous and the reader would not be led to confusion or misinterpretation. On the other hand, the paper is full of acronyms (some not defined) which would make the paper tedious and opaque to a reader unfamiliar with the field. This is important since a reader familiar with the field would not find much which is novel in the paper.

Lines 22 to 29 on page 10 are repeated as lines 30 to 4 (on page 11). The resolution of the maps in figures 5 to 16, particularly (g), needs improvement as the resolution only marginally allows the reader to see sufficient detail.

---

## Author Comment (AC1) · 19 Mar 2017

The authors would like to thank the editor and the reviewers for their precious time and invaluable comments. The corresponding changes and refinements are highlighted in yellow in the revised paper. Both authors' responses and revised manuscript are attached as in a PDF file (supplemet) below. Brief responses are also found below.

Reviewer #1: I review this same paper in July/August 2016, and as it looks, only a few of my major and minor comments sent back to the authors at that time have been addressed. This paper presents the implementation of a convective initiation (CI) algorithm that operates on Himawari-8 AHI data, with applications to the Korean Peninsula. Overall, a major concern related to the requirement that journals present NEW, innovative work is that this paper repeats much of the analysis methods in two papers, Mecikalski et al. (2010) and Mecikalski et al. (2015). If this journal is o.k. with the "newness" being that prior methods are developed using new (i.e. Himawari-8) data and not new methods, then the paper is not a bad presentation. But, the paper suffers from considerable grammatical problems. It need to be read by an English speaking person.

In the Mecikalski et al. (2010) paper, several Meteosat Second Generation (MSG) satellite based fields were defined outward of principal component and other information content analysis (that began with an assessment of many possible interest fields) for their value at predicting CI in the coming 1 hour. These are the results as stated in their Table 3. In the Mecikalski et al. (2015) paper, which defines the current GOES-R CI algorithm (albeit the authors have some confusion as to the authors/developers of the GOES-R CI algorithmâËŸAËĞ Tsee corrections below), the RF and LR machine learning approaches were applied to a reduced set of GOES-specific satellite predictor fields and also to NWP fields. Hence, this paper seemingly combined the Mecikalski et al. (2010) and Mecikalski et al. (2015) studies, with little new information, insights or analysis being done. I therefore again am not sure how appropriate it is to publish such a study that just re-applies already-published ideas, yet the authors have properly cited the relevant prior research and that the results are similar to these prior works. I reiterate that it may not be appropriate to publish results which are effectively _95%+ stated in other papers, based on the Methodology of Section 3, and the results in Tables 9 and 10 are also nearly the same for the RF and LR models as shown in Mecikalski et al. (2015). These authors have not done a complete analysis of all possible AHI datasets with respect to CI, which is in fact an ongoing activity in my research group today, and therefore do not shed new light on the value of all 13 infrared and the visible channels for predicting CI. I am therefore inclined to Reject this paper since it effectively duplicates prior work, while not significantly advancing our understanding.

–> Thanks for the comments. We agree that the present study used similar machine

learning approaches that were adopted in recent papers (Mecikalski et al., 2015; Han et al., 2015) to detect CI. Mecikalski et al. (2010) evaluated and analyzed Meteosat Second Generation (MSG) satellite-based interest fields using Principal Component Analysis (PCA) and Mecikalski et al. (2015) developed CI detection models based on random forest and logistic regression using GOES satellite and NWP model data. Our previous study (Han et al., 2015) also used decision trees and random forest to detect CI from COMS satellite data. However, we would like to say that the novelty of our present study when compared to the previous studies lies in the following two points: 1) Our present study is, as we know of, the first paper that evaluated Himawari-8 AHI data for CI detection. In our study, we solely focused on using AHI channel data without any ancillary data to detect CI for an operational purpose. While CI detection research has been widely conducted over US and Europe, it has had minimum exploration over Northeast Asia. This present study can contribute to the forecast and mitigation of heavy rainfall in Northeast Asia, especially during the rainy season (i.e., summer). 2) Our proposed machine learning-based approaches contain two new post processesâ Ăămajority voting and region growing, which are included in the revision. Since pixel-based CI detection is known to often result in salt-and-pepper noise and non-compact CI output, our proposed approaches include the post-processing to minimize such problems. The post-processing generally resulted in an increase of POD and a decrease of FAR.

We would like the reviewer to look at our fully revised manuscript attached. We significantly revised our manuscript according to your comments and those from the other reviewer. We improved our approaches by incorporating two post-processing techniques and added five additional validation cases (i.e., a total of 8 validation datasets) with more discussion to improve the quality of our study. Figures were updated with more clarity. Although it is not possible to directly compare our results to others' as different input and reference data were used, this present study showed good results comparable with Mecikalski et al. (2015). This implies that Himawari-8 satellite data (or future weather satellites with similar/more advanced specifications such as GOES-

R and GK-2A) can be solely used to detect CI, which enables the development of operational CI detection algorithms with high POD and low FAR. However, as shown in Mecikalski et al. (2015), model results such as convective available potential energy (CAPE), convective inhibition (CIN), and vertical shear (0-6km) can be effectively used to reduce FAR in the proposed CI detection algorithms.

If the Editor deems it appropriate to publish these results, then: (1) The authors need to carefully define all acronyms before they are used. This problem begins in the abstract and continues well into the paper. There are numerous spelling and grammatical issues (e.g., Himawari-8 is not capitalized on page 4/top, and the correct acronym is "EUMETSAT" on "EUMESAT" on page 2).

–> Thank you for your comments. We thoroughly checked acronyms and spelling problems from abstract to conclusion. A professional editing service was also used to improve the readability of the manuscript.

(3) Overall, it is not good to validate a CI algorithm using lightning observations, especially in the algorithm development stages. The authors do note the reason for poorer model performance when lightning data were used, later in the paper, as not all CI events go on to make lightning.

–> We agree that lightning observations are not appropriate to validate CI algorithms. As you said, clouds with heavy rainfall without lightning observations sometimes occurred. We added this explanation on page 12, lines 13 – 15. However, it might be useful where ground radar data are not available (e.g., ocean).

(4) Methodology/Section 3: Please remove the sentence "However, as mentioned, the GOES-R CI algorithm uses simple threshold values associated with the interest fields and the values were determined through many experimental simulations in a subjective way." First, the present GOES-R CI algorithm (Mecikalski et al. 2015) does NOT use "simple threshold values". Second, all prior research was NOT subjective in nature, but rather examined growing cumulus clouds in advance of CI with respect to physical

processes (cloud growth rates, updraft size/width, glaciation, cloud altitude, updraft longevity, etc.) as measured/observed by geostationary satellite infrared and visible datasets. Specifically, the Mecikalski et al. (2010) paper using MSG data was very focused on gaining understanding on how the interest fields behaved and subsequently how specific "threshold" values could be set, similar to how the original Mecikalski and Bedka (2006) study was performed. The way this sentence reads is that the prior work was just done without much thought, which was hardly the case.

–> Thank you for this comment. We are sorry that we thought that GOES-R CI algorithm is from National Oceanic and Atmospheric Administration (NOAA) National Environmental Satellite Data, and Information Service (NESDIS) center for satellite applications and research Algorithm Theoretical Basis Document (ATBD), Convective Initiation Version 2 (we think it is an old version), which we used in our original manuscript. We removed the sentence as suggested. We would like to say that our study is based on the research findings from the previous research papers including those the reviewer mentioned.

For more MINOR comments, I again suggest that the paper be reviewed and edited by an English-speaking person prior to acceptance.

–> We carefully proofread the manuscript several times. In addition, a professional editing service was used to improve the clarity and readability of the manuscript.

Please also note the supplement to this comment:
http://www.atmos-meas-tech-discuss.net/amt-2016-308/amt-2016-308-AC1-supplement.pdf

---

## Author Comment (AC2)

**Authors' responses (Agricultural and Forest Meteorology)**

The authors would like to thank the editor and the reviewers for their precious time and invaluable comments. ==The corresponding changes and refinements are highlighted in yellow in the revised paper== and are also summarized in our responses below. Authors' responses are in blue. Reviewer's comments are in black. When the manuscript is cited, it is shown in italics.

**Reviewer #2:**

This paper presents studies related to the use and selection of a convective initiation (CI) algorithm for application to Himawari-8 AHI data, specifically collected for the Korean Peninsula. The paper addresses questions within the scope of AMT although it does not introduce new concepts or ideas. It reaches interesting conclusions in the context of applying the data to the Korean Peninsula and, although the novelty of the paper is minimal, it gives a reasonable description of the issues involved with detecting CI. Publication of such analyses is not unusual for new instrumentation as it assists others and it provides a benchmark in the analysis process. In this case, the data analysis is limited in the confidence which we can have by the small number of days (in the figures) for which the results of the training data are applied. If the algorithms are truly to be 'validated over Northeast Asia', we need a better (larger) validation data-set. The English language in the paper would benefit from the advice of a native English speaker but it is not disastrous and the reader would not be led to confusion or misinterpretation.

➔ Thank you for your comments. We added five (5) more CI events for validation during June to August 2015-2016 because CI models were developed for the summer season in 2015. A total of validation datasets were eight (8), which we think reasonable when compared to previous CI studies (Mecikalski et al. 2006; Mecikalski et al. 2008; Walker et al. 2012; Merk and Zinner. 2013; Mecikalski et al. 2015).

We are developing seasonal CI models, which is the main topic of our next research paper. As Himawari-8 is relatively new, it takes time to get sufficient training samples for CI detection models for different seasons.

English was carefully revised. We also used a professional editing service to improve the clarity and readability of the manuscript.

The novelty of our present study when compared to the previous studies lies in the following two points: 1) Our present study is, as we know of, the first paper that evaluated Himawari-8 AHI data for CI detection. In our study, we solely focused on using AHI channel data without any ancillary data to detect CI for an operational purpose. While CI detection research has been widely conducted over US and Europe, it has had minimum exploration over Northeast Asia. This present study can contribute to the forecast and mitigation of heavy rainfall in Northeast Asia, especially during the rainy season (i.e., summer). 2) Our proposed machine learning-based approaches contain two new post processes—majority voting and region growing, which are included in the revision. Since pixel-based CI detection is known to often result in salt-and-pepper noise and non-compact CI output, our proposed approaches include the post-processing to minimize such problems. The post-processing generally resulted in an increase of POD and a decrease of FAR.

**Authors' responses (Agricultural and Forest Meteorology)**

We would like the reviewer to look at our fully revised manuscript attached. We significantly revised our manuscript according to your comments and those from the other reviewer. We improved our approaches by incorporating two post-processing techniques and added five additional validation cases (i.e., a total of 8 validation datasets) with more discussion to improve the quality of our study. Figures were updated with more clarity. Although it is not possible to directly compare our results to others' as different input and reference data were used, this present study showed good results comparable with Mecikalski et al. (2015). This implies that Himawari-8 satellite data (or future weather satellites with similar/more advanced specifications such as GOES-R and GK-2A) can be solely used to detect CI, which enables the development of operational CI detection algorithms with high POD and low FAR. However, as shown in Mecikalski et al. (2015), model results such as convective available potential energy (CAPE), convective inhibition (CIN), and vertical shear (0-6km) can be effectively used to reduce FAR in the proposed CI detection algorithms.

On the other hand, the paper is full of acronyms (some not defined) which would make the paper tedious and opaque to a reader unfamiliar with the field. This is important since a reader familiar with the field would not find much which is novel in the paper.

➔ Thank you for your comments. We thoroughly checked acronyms from abstract to conclusion

Lines 22 to 29 on page 10 are repeated as lines 30 to 4 (on page 11). The resolution of the maps in figures 5 to 16, particularly (g), needs improvement as the resolution only marginally allows the reader to see sufficient detail.

➔ Thank you for your comments. We removed the repeated paragraph. Most figures were updated with new results. Resolution was also improved.

[revised manuscript text omitted]

**3.3 Post-processing of CI cloud objects**

Similar to the existing studies (citations), pixel-based CI detection models were developed in this study. However, pixel-based CI detection has some drawbacks. First, salt-and-pepper noise with one to three pixels often occurs, which is not related to CI cloud clusters. Second, although CI clouds are typically compact, non-compact CI cloud objects are sometimes detected due to rough cloud tops. In order to minimize these problems, two techniques were adopted—majority filtering and region growing. We tested different window sizes (from 2 to 5 pixels) for majority filtering, and determined a 2x2 window as an optimum size based on visual inspection of the resultant CI cloud objects. If there were only one or two CI pixels in the 2x2 window, the CI pixels were excluded. Otherwise (i.e., more than two CI pixels), all pixels were considered as CI pixels. After the majority filtering, region growing was conducted to make the detected CI cloud objects more compact and aggregated. Region growing has been widely used to segment images to produce objects (i.e., homogeneous regions) in the field of remote sensing. The basic concept of region growing is to examine neighboring pixels from seed pixels and determine whether they should be added to the region of a seed (citations). In this study, CI pixels were designated as seed points and $T_B$ at 11.2 μm was used as a background field to examine the homogeneity of regions. Regions grow while the difference between the temperature averaged within a region and temperature at a neighboring pixel is less than 0.5, which is

empirically determined. This post-processing was conducted to reduce salt-and-pepper noise and false alarm rates (FAR) for CI detection.

**4 Performance and validation of CI detection models**

**4.1 Performance of CI detection models with post-processing**

[revised manuscript text omitted]

Fig. 4 shows CI detection results before and after the two post-processes. Incorrectly detected small CI pixels (Figs. 4a, 4c) were removed through the majority voting. In addition, the CI objects had a more compact shape with few holes (Figs. 4b,

20   4d) through the region growing. This post-processing resulted in decreasing FAR and increasing POD by making CI objects grow and merge, which is discussed in the next sub-section.

Fig. 5 depicts the accuracy metrics before and after the post-processing—majority voting and region growing. The figure clearly shows that the post-processing led to an increase of POD and decrease of FAR. In particular, FAR decreased about 4% after the post-processing was applied when lightning data was used for validation. Higher POD and lower FAR resulted in

25   higher OA and CSI, which implies that the proposed post-processing was effectively used to improve the performance of the CI detection models.

**4.2 Validation of three CI cases with ground radar and lightning data.**

The four CI detection models were applied to the eight cases of CI events over the Korean Peninsula and validated using two types of reference datasets (i.e., weather radar and lightning observations). Figs. 6 and 7 show the validation metrics of the

30   models based on each reference dataset. Fig. 8 represents averaged lead time for each validation by model.

Overall, DT produced the highest POD values regardless of the reference data used (i.e., for both lightning and ground radar). However, it over-predicted CI objects, which resulted in high FAR. Although RF yielded slightly lower POD than DT when

lightning data was used, RF showed much lower FAR than DT, which led to higher OA and CSI than DT (Fig. 6). On the other hand, when ground radar was used for validation, since FAR was calculated considering the effective radius of the radar, RF produced FAR similar to that of DT, which resulted in higher OA for DT than RF (Fig. 7).

There is not much difference in FAR values between DT and RF when radar reference data were used. However, CI objects produced from DT tended to be largely scattered through visual inspection of the results, which may result in confusion to the users of CI product (i.e., forecasters). For probabilistic RF and LR models, 50% was used as a threshold to detect CI. Both produced relatively lower FAR and POD values than DT. Although the threshold to detect CI can be optimized for probabilistic RF and LR models to improve the performance, it is beyond the scope of this research. Considering both results used lightning and ground radar data, RF appeared to be the best CI detection model, which can be confirmed by the averaged CSI values (Figs. 6 and 7). Since lead time is calculated based on hits, DT mostly produced the longer averaged lead time than the other three models (Fig. 8). However, when radar data was used for validation, the difference in averaged lead time between DT and RF was less than 1 min.

Mecikalski et al. (2015) used 50% as the threshold to identify CI from the results of probabilistic RF and LR and showed results similar to ours depicted in Fig. 7, i.e., a slightly better performance by probabilistic RF than LR. Unlike Mecikalski et al. (2015) that used both 9 channels of GOES satellite data and 16 NWP model data, this present study solely focused on using satellite data—Himawari-8 AHI channels. Although it is not possible to directly compare our results to others' as different input and reference data were used, this present study showed good results comparable with Mecikalski et al. (2015). This implies that Himawari-8 satellite data (or future weather satellites with similar/more advanced specifications) can be solely used to detect CI, which enables the development of operational CI detection algorithms with high POD and low FAR. However, model results such as convective available potential energy (CAPE), convective inhibition (CIN), and vertical shear (0-6km) can be effectively used to reduce FAR in the proposed CI detection algorithms.

Fig. 9 shows CI areas for the case of CI events on 7th August 2015 07:50 (UTC) predicted by the DT, deterministic RF, probabilistic RF, and LR models, respectively. All models showed better performance in terms of FAR, OA, and CSI based on the weather radar observations rather than the lightning observations. This is because the number of CI objects detected by the radar is smaller than that by the lightning observations due to the limited effective radius, similar to the case on 12th June. The DT and deterministic RF models detected CI areas around the Northwestern Korean peninsula. Such predicted CI areas might be correct despite the lack of lightning observations, but the weather radar data from China Meteorological Administration (CMA) has not been available over the region and hence we were not able to confirm whether the CI objects were correctly identified. CI events on 30 June 2016 08:40 (UTC) were depicted in Fig. 10. The DT and LR models over-detected CI objects, which resulted in high FAR. While FAR based on radar data was much lower than that based on lightning data, the averaged lead time based on lightning data was longer than that based on radar data because there were missing CI objects in radar echoes above 35 dBZ around the latitude of 38°N and longitude of 128.5°E.

CI occurrence on 6th July 2016 05:30 (UTC) was shown in Fig. 11. All models were not able to detect CI events which occurred around the west coast of Korean Peninsula, even at the same time as CI occurred. The CI objects located around the

west coast had a relatively high temperature at 11.2 μm. It is reported that warm-type heavy rainfall, lower storm height, with lower ice content have often developed over the ocean in East Asia (Sohn et al. 2013; Song and Sohn, 2015). Considering the temperature range of the CI objects, they appeared to be warm-type heavy rainfall clouds. As all four models are empirical, the missed CI cases imply that the training data didn't contain such warm-type CI clouds. Fig. 12 shows CI events on 24th July 2016 14:50 (UTC) generated by four models. No lightning data was available in this case. CI objects were detected around latitude 38°N and longitude 128°E by all models except for the probabilistic RF model. Since CI objects in this case rapidly grew, it made lead time short and early detection of CI difficult. While FAR was almost zero, the averaged lead time was less than 20 min (Figs. 7 and 8).

The two validation datasets, i.e., the weather radar and lightning observations, influenced the assessment of model performance. Since the weather radar sites are located inland, convective clouds over the ocean were out of the detection radius and therefore less likely to be detected. Meanwhile, lightning observations can even detect CI objects over the distant sea, but it is hard to identify the exact location of lightning in clustered clouds. These limitations in each verification dataset provide uncertainty in estimating the actual forecast skill of the CI detection models. Furthermore, since there were some heavy rainfall clouds without lightning observations, lighting data for CI reference was not reliable in the algorithm development stage.

Due to the similar number of hits from the four models, the lead time of all four models was around 32 to 40 min. This indicates that CI over the Korean Peninsula can be forecasted using the Himawari-8 AHI images with a usable lead time of 30 to 40 min, which is reasonably comparable to the lead time for CI detection (~30–45 min) in the literature (Han et al., 2015; Mecikalski et al., 2015). AHI images 50 min before CI occurrence were used to detect CI in this study. If the AHI data collected a few hours before CI occurrence was used in the development of CI detection models, a longer lead time could possibly be achieved (refer to the supplementary material as an example). Additionally, a rapid scan mode with 2 min temporal resolution may be used for rapidly growing clouds.

[revised manuscript text omitted]

**Figure 4.** Before and after post-processing of CI cloud objects using majority voting and region growing. (a) An example of deterministic CI detection generated by DT before post-processing. (b) Deterministic CI detection generated by DT after post-processing. (c) Deterministic CI detection generated by RF before post-processing. (d) Deterministic CI detection generated by RF after post-processing. While dashed white circles show that region growing made CI objects more compact with few holes, dashed yellow circles show that majority voting effectively removed salt-and-pepper noise.

[Figure]

**Figure 5.** Quantitative assessment based on deterministic RF before and after post-processing. (a) validation metrics of RF using lightning data before post-processing. (b) validation metrics of RF using lightning data after post-processing. (c) validation metrics of RF using radar CAPPI data before post-processing. (d) validation metrics of RF using radar CAPPI data after post-processing. The values next to boxes correspond to the average metric values in percentage.

[Figure]

Figure 6. Validation metrics based on lightning data for the (a) deterministic DT and (b) RF, (c) probabilistic RF, and (d) LR models. Boxplots of validation metrics based on lightning data for (e) POD, (f) FAR, (g) OA, and (h) CSI. While red lines in the boxes represent mean values, grey lines represent median values.

[Figure]

**Figure 7.** Validation metrics based on radar CAPPI data for the (a) deterministic DT and (b) RF, (c) probabilistic RF, and (d) LR models. Boxplots of validation metrics based on radar CAPPI data for (e) POD, (f) FAR, (g) OA, and (h) CSI. While red lines in the boxes represent mean values, grey lines represent median values.

[Figure]

**Figure 8.** Averaged lead time based on lightning and radar CAPPI data for the deterministic DT and RF, probabilistic RF, and LR models. Box plot of lead time based on lightning and radar CAPPI data for the (c) lead time based on lightning data and (d) lead time based on radar CAPPI data. While red lines represent in the boxes mean value, grey lines represent median value.

[Figure]

**Figure 9.** Deterministic CI detection map derived by DT, RF, probabilistic RF, and LR on 7 August 2015 07:30 (UTC): (a) DT, (b) RF, (c) Prob RF, and (d) LR 30 min before CI occurrence. (e) is the 1.5 km radar CAPPI image at 08:00 (UTC). While predicted CI is in red, the locations of lightning occurrences at 08:00 (UTC) are presented in yellow dots. The radar echoes above 35 dBZ beyond the effective radius of radar are shaded with red hatch.

[Figure]

**Figure 10.** Deterministic CI detection map derived by DT, RF, probabilistic RF, and LR on 30 June 2016 08:00 (UTC): (a) DT, (b) RF, (c) probabilistic RF, and (d) LR 40 min before CI occurrence. (e) is the 1.5 km radar CAPPI image at 08:40 (UTC). While predicted CI is in red, the locations of lightning occurrences at 08:00 (UTC) are presented in yellow dots.

[Figure]

**Figure 11.** Deterministic CI detection map derived by DT, RF, probabilistic RF, and LR on 6 July 2016 05:30 (UTC): (a) DT, (b) RF, (c) probabilistic RF, and (d) LR 40 min before CI occurrence. (e) is the 1.5 km radar CAPPI image at 06:10 (UTC). While predicted CI is in red, the locations of lightning occurrences at 06:10 (UTC) are presented in yellow dots. The radar echoes above 35 dBZ beyond the effective radius of radar are shaded with red hatch.

[Figure]

**Figure 12.** Deterministic CI detection map derived by DT, RF, probabilistic RF, and LR on 24 July 2016 14:50 (UTC): (a) DT, (b) RF, (c) probabilistic RF, and (d) LR 10 min before CI occurrence. (e) is the 1.5 km radar CAPPI image at 15:00 (UTC). The radar echoes above 35 dBZ beyond the effective radius of radar are shaded with red hatch.